# Chronic intracranial recordings in the globus pallidus reveal circadian rhythms in Parkinson's disease

Jackson N. Cagle [1,2], Tiberio de Araujo[2], Kara A. Johnson [1,2], John Yu [1,2], Lauren Fanty[1,2], Filipe P. Sarmento [2], Simon Little [3], Michael S. Okun [1,2], Joshua K. Wong[1,2] & Coralie de Hemptinne [1,2] ✉

Circadian rhythms have been shown in the subthalamic nucleus (STN) in Parkinson's disease (PD), but only a few studies have focused on the globus pallidus internus (GPi). This retrospective study investigates GPi circadian rhythms in a large cohort of subjects with PD (130 recordings from 93 subjects) with GPi activity chronically recorded in their home environment. We found a significant change in GPi activity between daytime and nighttime in most subjects (82.4%), with a reduction in GPi activity at nighttime in 56.2% of recordings and an increase in activity in 26.2%. GPi activity in higher frequency bands (>20 Hz) was more likely to decrease at night and in patients taking extended-release levodopa medication. Our results suggest that circadian fluctuations in the GPi vary across individuals and that increased power at night might be due to the reemergence of pathological neural activity. These findings should be considered to ensure successful implementation of adaptive neurostimulation paradigms in the real-world.

Parkinson's disease (PD) is a neurodegenerative movement disorder characterized by the loss of dopaminergic neurons, resulting in the development of both motor and nonmotor symptoms. Deep brain stimulation (DBS) is a safe and effective neuromodulation therapy for medication-refractory motor symptoms in select PD patients. This invasive neurosurgical procedure involves delivering electrical stimulation to deep brain nuclei through electrodes implanted in the basal ganglia. In PD, DBS is commonly targeted to either the globus pallidus internus (GPi) or the subthalamic nucleus (STN), both of which are pivotal nodes in the basal ganglia–thalamo–cortical networks underlying PD pathophysiology[1].

Circadian rhythms are endogenous fluctuations that orchestrate a wide range of complex physiological and behavioral processes across the ~24-h sleep/wake cycle[2]. Dopamine contributes to sleep/wake cycle regulation[3–5] via the hypothalamus and mesolimbic pathways[6–9]. Circadian rhythm disturbances have been observed in PD, resulting in a broad range of symptoms, including sleep-wake disorders, autonomic dysregulation, temperature imbalance, and

motor fluctuations[10]. Recent studies have revealed that the neural activity in the GPi and STN in subjects with PD exhibited circadian patterns[11] and fluctuated during various stages of sleep[12–20]. A recent study evaluating chronic STN local field potential (LFP) recordings in the home environment has revealed beta power circadian fluctuations, with a consistent reduction of beta power during the overnight hours[12]. This result supported prior findings from acute STN recordings performed in clinic using externalized electrodes over one night, which showed that on average beta (13–30 Hz) and gamma (30–60 Hz) power in the STN decreased, while delta (0–3 Hz), theta (3–7 Hz), and alpha (7–13 Hz) power increased during non-rapid-eye movement (NREM) sleep when compared to wakefulness and rapid-eye movement (REM) sleep[13–15,18,19]. While studies investigating circadian rhythms via subcortical electrophysiological signals have largely focused on STN, two recent studies by Yin et al. have focused on GPi recordings in PD. When compared to Huntington's disease and dystonia, Yin et al. found that similar to STN, pallidal beta and gamma activity was reduced in NREM compared to

[1]Department of Neurology, University of Florida, Gainesville, FL, USA. [2]Norman Fixel Institute for Neurological Diseases, University of Florida, Gainesville, FL, USA. [3]Department of Neurology, University of California San Francisco, San Francisco, CA, USA. ✉e-mail: Coralie.deHemptinne@neurology.ufl.edu

REM sleep and the awake state, while delta, theta, and alpha power were increased during NREM sleep[16,17].

Recent research has increasingly focused on the development of adaptive DBS (aDBS) algorithms, which use neural signals as feedback control to increase or decrease stimulation based on the presence of pathological activity[21]. However, natural fluctuations of biomarkers due to sleep/wake cycles as described by previous studies may add complexity to the implementation of aDBS. Chronic brain recordings have shed light on rhythmic activity, more specifically circadian rhythms, that could potentially alter the control signals of aDBS algorithms[11].

Although the presence of circadian rhythms in the basal ganglia has been investigated, most studies have been conducted in small cohorts and have focused on STN activity recorded in the off-medication state through externalized electrodes in controlled environments. Therefore, how circadian rhythms could modulate basal ganglia activity, in particular in the GPi, in the home environment, and under naturalistic conditions, remains largely unknown. The objective of this study was to characterize circadian patterns of neural activity recorded in the GPi using longitudinal, at-home recordings under naturalistic conditions (including unrestricted therapeutic medications and chronic stimulation conditions) via a sensing-enabled chronically implanted neurostimulator in a large retrospective cohort of subjects with PD.

## Results

This retrospective study includes 93 subjects with PD (130 unique hemispheres with recordings) from the UF INFORM database who met the inclusion criteria. Of these, 72 were males, and 21 were females (77.4% and 22.6%, respectively), with an average age at surgery of 68.3 ± 9.4 (mean ± std) years and an average disease duration of 14.1 ± 5.9 (mean ± std) years. Subjects were implanted with the Medtronic Percept PC neurostimulator attached to unilateral ($n = 38$, 40.9%) or bilateral ($n = 55$, 59.1%) DBS leads in the GPi. Among them, 58 hemispheres (44.6%) were implanted with Medtronic 3387 quadripolar DBS electrodes, and 72 hemispheres (55.4%) were implanted with Medtronic SenSight segmented DBS electrodes (B33015). Table 1 summarizes the characteristics of the subjects in the cohort.

In each subject, neural activity was chronically recorded in their home environment under both therapeutic chronic medication and stimulation conditions. To ensure data consistency, we selected the first 5 consecutive days (120 h) of chronic sensing without changes in recording and stimulation settings for each subject and hemisphere. On average, the recordings occurred 59.9 ± 129 (mean ± std) days after pulse generator device implantation, which occurs 4 weeks after lead implantation surgery. The frequency of oscillatory activity sensed chronically was selected in the off-medication state by the clinicians during a DBS programming session. The clinicians typically selected the most prominent peak on the power spectral density generated after a short survey recording. The sensing frequency ranged from 7.81 Hz to 25.39 Hz across the cohort. For analysis, sensing frequencies were grouped in canonical frequency bands defined as alpha (7–12 Hz), low beta ( >12 and ≤20 Hz), and high beta ( >20 and ≤30 Hz). The alpha band and low beta band were more commonly recorded (alpha $n = 55$, 42.3%; low beta $n = 52$, 40.0%), compared to high beta ($n = 23$, 17.7%). Most subjects were receiving continuous high-frequency stimulation therapy during the chronic recordings. All subjects were programmed in a monopolar configuration and the two contacts adjacent to the contact(s) used for therapeutic stimulation were selected for bipolar chronic sensing. Although stimulation settings were constant within subjects, therapy settings varied across subjects, with stimulation pulse widths ranging from 60 to 120 μsec, frequencies ranging from 125 Hz to 185 Hz, and amplitudes ranging from 0 mA to 4.5 mA.

Data were collected in subjects implanted in STN, using the same inclusion criteria (23 subjects, 30 unique hemispheres). The same analyses were performed as control and are provided in the supplementary information for comparison.

### GPi power mainly decreases at nighttime but may sometimes increase

Figure 1A shows chronic GPi high beta (24.41 ± 2.50 Hz) activity recorded in an individual subject over five consecutive days. Power fluctuations were observed with a consistent reduction during nighttime. The power reduction manifested around midnight. Vertical lines indicate the time of the events marked (suggesting the subject was awake), and ranged from 7 am to midnight (EDT), corresponding to a period of higher beta power during the daytime period. The polar plots illustrate the circadian rhythms, showing that beta power was higher during the day (when the events were marked). Interestingly, increased power at night was also observed in other subjects. Figure 1B shows an example of low beta (19.53 ± 2.50 Hz) activity in a different subject with GPi recordings exhibiting consistently increased power at night and decreased during the day when the subject marked events.

For group analyses, the neural power was z-score normalized and averaged over daytime (3:00–8:00 pm) and nighttime (0:00–5:00 am). Figure 1C shows the GPi normalized power ($n = 130$ hemispheres) plotted over 24 h, sorted by the highest power in the daytime, and grouped in the main canonical spectral band (alpha, low beta, and high beta). Visual inspection shows the presence of a strong circadian rhythm in most recordings. We found increased power at nighttime in 26.2% of all recordings (34 hemispheres) while decreased power was observed in 56.2% (73 hemispheres). In 23 hemispheres (17.7%), there was no statistically significant difference between daytime and nighttime power distributions ($t$-test, $p_{corrected} > 0.05$). When analyzing the beta band (13–30 Hz) separately, we found that 16.0% (12 hemispheres) showed an increase in power at nighttime, 69.3% (52 hemispheres) showed a decrease in power at nighttime, and 14.7% (11 hemispheres) of the recordings had no statistically significant difference in power between daytime and nighttime ($t$-test, $p_{corrected} > 0.05$).

### GPi circadian rhythms differ across frequency bands

Considering the variability in circadian rhythms, characterized by both increase and decrease in power at nighttime, we investigated the impact of sensing frequency on the variability. For each recording, the change in day and night normalized power was calculated and plotted against the center of the sensing frequency (Fig. 2A). A one-way analysis of variance (ANOVA) to compare across frequency bands revealed a significant effect of sensing frequency in GPi on daytime/nighttime power change ($F = 18.22$, $p_{corrected} < 0.001$). Post-hoc analysis with Tukey's test indicated that the alpha band power showed a greater increase at nighttime than low beta ($q = −0.47$, $p_{corrected} = 0.004$) and high beta ($q = −1.09$, $p_{corrected} < 0.001$), and low beta showed a greater increase at nighttime than high beta ($q = 0.63$, $p_{corrected} = 0.003$) (Fig. 2B). Comparison within each frequency band showed a statistically significant daytime/nighttime change in power in both low beta (mean=0.38, $t = 3.43$, $p_{corrected} = 0.004$) and high beta (mean=1.01, $t = 9.08$, $p_{corrected} < 0.001$). Averaged positive values indicate that power in low beta and high beta bands was more likely to decrease at night. In contrast, alpha power was equally decreased or increased at night, as indicated by a mean day/night change non-significantly different from 0 (mean = −0.04, $t = −0.36$, $p_{corrected} = 1.0$). These results suggest that the direction of GPi circadian rhythm depends on the frequency bands sensed, with a decrease in power at nighttime observed in all frequency bands and an increase in power at nighttime in mainly the alpha and low beta bands.

## Extended-release levodopa modulates GPi circadian rhythms

Medications, stimulation, and the patients' symptoms are known to modulate basal ganglia activity in PD, especially in the beta band. Therefore, a general linear model (GLM) was built to test the effects of stimulation parameters, medications, and motor phenotype on the beta power (13–30 Hz) circadian rhythms in GPi. In particular, we investigated whether the change in beta power (dependent variable) between daytime and nighttime was significantly modulated by the following independent variables: the levodopa equivalent daily dose (LEDD)[22,23], the use of extended-release levodopa at nighttime, the use of non-dopaminergic medication at nighttime, the location of sensing (ventral vs. dorsal), the total electrical energy delivered (TEED), the Unified Parkinson's disease rating scale (UPDRS) total scores, and the subject's motor phenotype (tremor, intermediate, or akinetic subtype). The PD motor phenotype, LEDD, and TEED were calculated based on established methods[22–25]. The location of sensing was determined as the mid-point between the 2 contacts used for sensing, which corresponded to the contact used for stimulation and was defined as ventral (stimulation contact 1) or dorsal (stimulation contact 2). The locations of all subjects' stimulating contacts are visualized in Fig. 3. Sensing locations in subjects with increased power at nighttime (blue) and decreased power at nighttime (red) did not show clear differences.

Table 2 summarizes the results of the GLM. The model indicated a significant effect of nighttime extended-release levodopa on beta power circadian rhythms ($z = 2.24$, $p_{corrected} = 0.039$) and the intercept term ($z = 2.97$, $p_{corrected} = 0.001$). All other variables were not significantly associated with the daytime/nighttime change in beta power. Figure 2C shows the distributions of subjects using or not using nighttime extended-release levodopa. A reduction of power at nighttime was observed in all subjects taking extended-release levodopa, except two. The intercept term indicates a significant bias

### Table 1 | Subject demographics, clinical characteristics, and recording settings

| GPi | |
|---|---|
| **Demographics** | |
| Total number of subjects | 93 (130 hemispheres) |
| Sex | M: 72 (76.3%) F: 21 (23.7%) |
| Age at surgery (years) | 66.0 ± 9.1 |
| Disease duration (years) | 11.7 ± 5.6 |
| **Symptoms and therapy characteristics** | |
| Preoperative UPDRS | 35.4 ± 12.5 |
| TEED (μJ) | 221 ± 214 |
| LEDD (mg) | 1275 ± 691 |
| **Recording settings** | |
| Recording durations (days) | 5 ± 0 |
| Time since implant (days) | 59.9 ± 129.0 |
| Sensing frequency (Hz) | 15.1 ± 4.6 |

*M* Male, *F* Female, *UPDRS* unified Parkinson"s disease rating scale, *TEED* total electrical energy delivered, *LEDD* levodopa equivalent daily dosage, *μJ* microjoule, *mg* milligram, *Hz* Hertz
All values are reported as mean ± standard deviation, except sex and total number of subjects/ hemispheres.

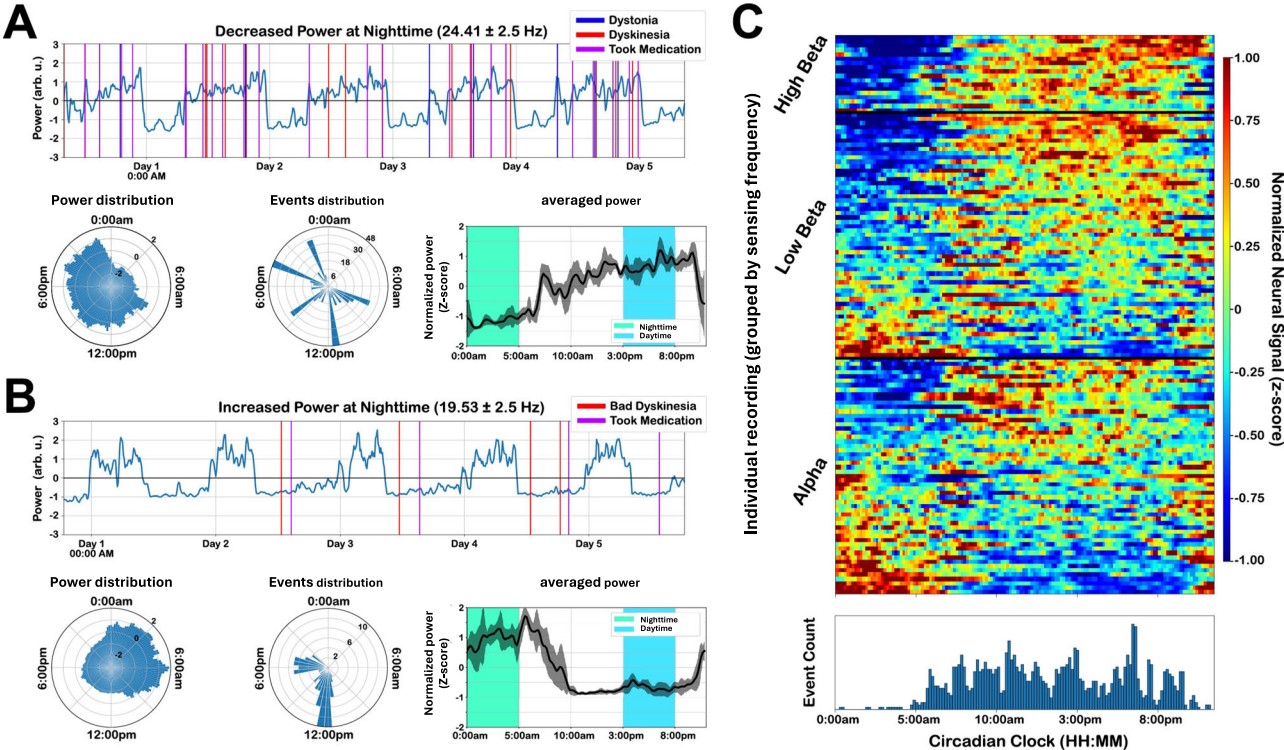

**Fig. 1 | Circadian rhythms in the GPi. A** Example subject exhibiting GPi beta band (24.41 ± 2.5 Hz) power decreasing at night. Pink vertical lines indicate the time of events marked by the subject and are used here as a surrogate marker of the awake/ sleep period. Circular polar plots show the beta power (left panel) and the number of events marked (middle panel) in a 24-h circadian clock with a 1-h averaged increment. The mean and standard error of the normalized spectral power averaged over a 24-h cycle is shown on the right panel. The green and blue shades indicate the nighttime and daytime periods, respectively, used for subsequent analysis (see methods). **B** Example subject exhibiting GPi beta band (19.53 ± 2.5 Hz) power increasing at night. Same convention as in panel A. **C** Circadian heatmap showing GPi power across all individual recordings (*n* = 130) normalized and plotted over a 24-h circadian clock. GPi power is sorted from the most decreased (blue) to the most increased (red) at night after grouping into canonical frequency bands. The number of events reported by all subjects over a 24-h circadian clock is shown below the circadian heatmap and used as a surrogate marker of the awake/sleep period. Hz Hertz, arbu arbitrary units; Source data is provided in the Source Data file.

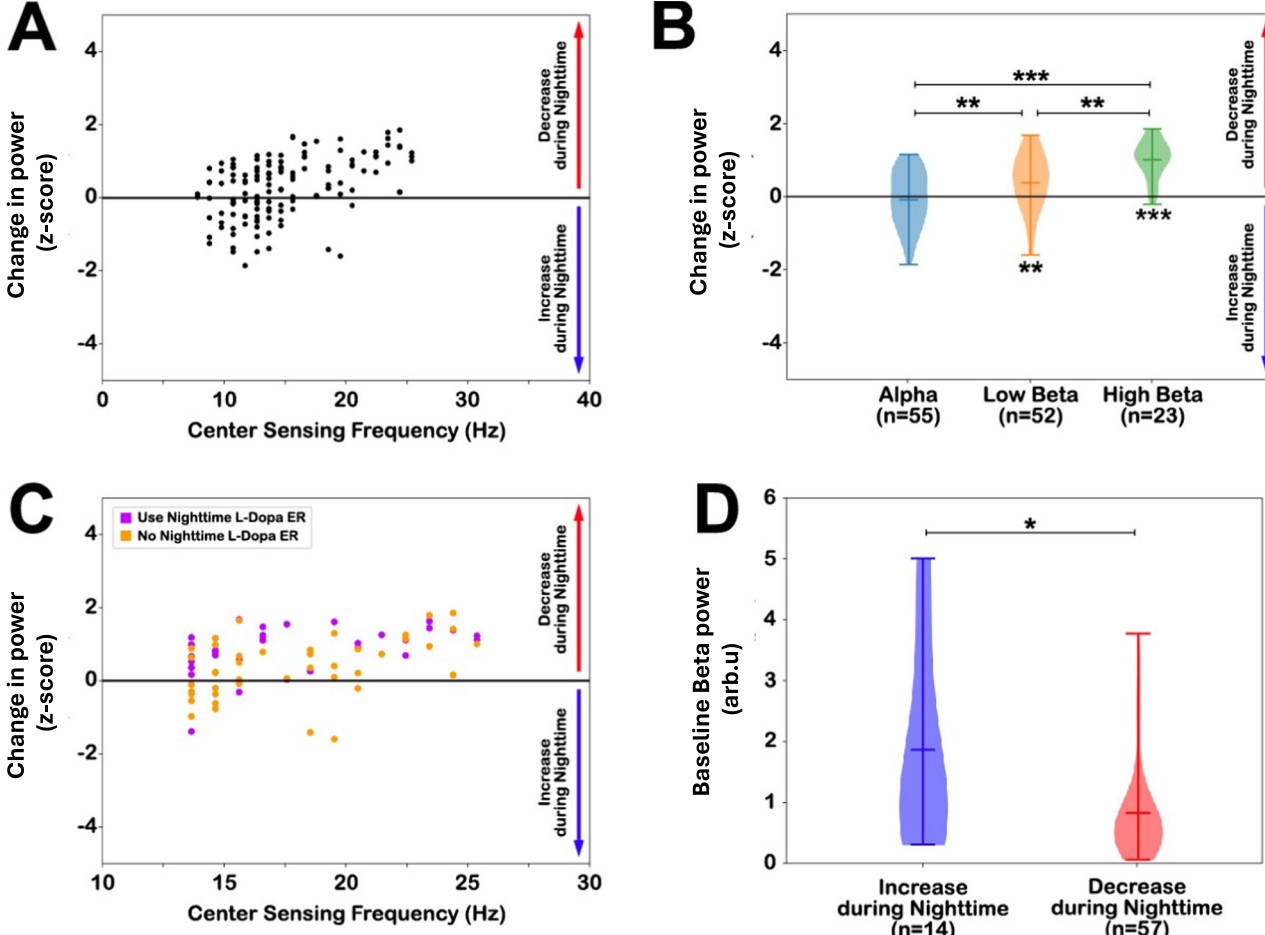

**Fig. 2 | Factors influencing circadian rhythms in the GPi. A** Scatter plot of the changes in spectral power between daytime and nighttime period (day–night), for each GPi hemisphere recording ($n = 130$), in relation to the center frequency of the recordings. A positive change indicates a decrease in power at nighttime and a negative change indicates an increase in power at nighttime. **B** Violin plot quantifying the changes in power during day and night for each canonical frequency band (mean and range); Alpha ($\leq 12$ Hz), low beta ($> 12$ and $\leq 20$ Hz), and high beta ($> 20$ and $\leq 30$ Hz). Low beta ($p_{corrected} = 0.004$) and high beta ($p < 0.001$) power was more decreased power at nighttime (indicated by positive values and *** below violins). At the same time, alpha power was not statistically different from 0 ($p_{corrected} = 1.0$), indicating that power was either increased or decreased at night. The high beta band was statistically different from the alpha band ($p_{corrected} < 0.001$) and low beta band ($p_{corrected} = 0.003$), and the low beta band was statistically different from the

alpha band ($p = 0.004$) using post-hoc Tukey's tests following a one-way ANOVA. **C** Scatter plot of circadian beta power ($> 12$ and $\leq 30$ Hz) in subjects treated with levodopa (L-Dopa) extended-release (ER) (purple) and those who do not take levodopa ER (orange). In subjects taking levodopa ER at nighttime, beta power was decreased more at night ($p_{corrected} = 0.002$) than in subjects who were not treated with levodopa ER. **D** Comparison of in-clinic off-medication off-stimulation awake baseline beta band power in subjects with increased and decreased power at nighttime (mean and standard error). There was a statistical difference between the two groups ($p_{corrected} = 0.028$; non-parametric Mann–Whitney $U$ test). (*) denotes $p < 0.05$, (**) denotes $p < 0.01$, and (***) denotes $p < 0.001$. All $p$ values were corrected for multiple comparisons with Bonferroni correction. $n$ number of recordings; Hz Hertz; L-Dopa ER levodopa extended-release, arbu arbitrary units. Source data is provided in the Source Data file.

toward a decrease in power at nighttime, which confirms that most subjects showed this trend.

The GLM results suggest that the increased power at night might be related to the re-emergence of pathological beta power when PD medications are wearing off at nighttime. Therefore, we retrieved the baseline in-clinic neural recordings acquired in the off-medication and off-stimulation conditions from subjects with beta band power chronically sensed. The baseline recordings were available in a subset of subjects (71 of 75 subjects with beta band power chronically sensed). We extracted the data recorded from the contact pair used for chronic sensing, computed the power spectral density, and averaged it over the beta band (13–30 Hz). Fourteen of the recordings (19.7%) showed increased power during nighttime and 57 (80.3%) showed decreased power during nighttime. A non-parametric Mann–Whitney $U$ test showed that subjects with increased beta power at nighttime had a significantly higher beta power at baseline compared to subjects with

decreased beta power at nighttime ($U = 229$, $p_{corrected} = 0.028$; Fig. 2D). These results suggest that beta power was more likely to increase at nighttime in subjects who were not taking extended-release levodopa and who had a large beta activity in the off-medication state. Therefore, in some cases, increased beta power at night may be associated with the re-emergence of pathological beta oscillations when the dopaminergic medication wears off at night.

## Discussion

This retrospective study characterizes the effect of circadian rhythms on PD GPi neurophysiology using a commercially available device in a large cohort under naturalistic therapy conditions in their home environment. The data revealed that 1) GPi is significantly modulated by circadian rhythms in most PD subjects, 2) although power was mainly reduced at night, increased power was also found in 26.2% of hemispheres, and 3) the frequency band sensed, the baseline power

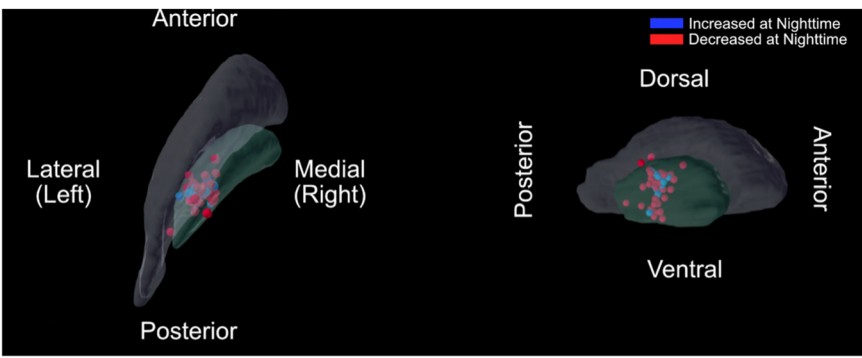

**Fig. 3 | Lead and sensing localizations.** Leads were localized in normalized atlas space (MNI152) in axial (left panel) and sagittal view (right panel). The segmentation of the globus pallidus internus (GPi) and externus (GPe) from the DISTAL atlas are shown in green and gray, respectively. The blue dots indicate the center of bipolar sensing contact pairs for subjects with increased beta power at nighttime and red dots for subjects with decreased beta power at nighttime. No cluster differentiation was found on visual inspection. Source data is provided in the Source Data file.

**Table 2 | Generalized linear model for the effect of stimulation therapy, medications, and motor symptoms on beta band (13–30 Hz) circadian rhythms**

|  | Coefficient | Std. Err. | z | p > |z| |
|---|---|---|---|---|
| **Intercept term** | 1.2846 | 0.384 | 3.342 | **0.001** |
| **Stimulation location (dorsal: 1, ventral: 0)** | −0.3737 | 0.194 | −1.924 | 0.054 |
| **TEED (μJ)** | 254.7574 | 420.308 | 0.606 | 0.544 |
| **LEDD (mg)** | −0.0001 | 0 | −0.882 | 0.378 |
| **Nighttime Levodopa ER (use: 1, not used: 0)** | 0.3946 | 0.191 | 2.062 | **0.039** |
| **Sleep medication (used: 1, not used: 0)** | −0.2965 | 0.201 | −1.475 | 0.14 |
| **PD subtype (tremor/akinetic ratio)** | 0.0769 | 0.121 | 0.637 | 0.524 |
| **UPDRS** | −0.0099 | 0.007 | −1.519 | 0.129 |

*ER* Extended-release, *TEED* total electrical energy delivered, *LEDD* levodopa equivalent daily dosage, *UPDRS* Unified Parkinson's Disease Rating Scale, *μJ* microjoule, *mg* milligram. Source data is provided in the Source Data file.

(off medication, off DBS), and the use of extended-release levodopa influenced the neurophysiological circadian rhythms.

Our findings revealed that GPi activity was strongly modulated by circadian rhythms, and the frequency of neural activity recorded may significantly affect the direction of the spectral power changes (increase or decrease) between day and night. Studies exploring circadian rhythms in STN LFP[11,12,14,15,26] have shown that different canonical frequency bands behave differently during sleep, with a general trend of slowing brain oscillations, shown by increased low-frequency power (< 12 Hz) and decreased beta and gamma power (> 13 Hz). Due to the lack of objective sleep stage measurements, we cannot conclude the effect of circadian rhythms during specific stages of sleep in this retrospective study. However, our results generally match the observations during the NREM stage found in previous studies. However, there was more variability within the alpha and low beta bands. The increase in power at night, especially in low beta power, could be explained by the large spectral window of the recording. Indeed, the neural power recorded chronically is set at a preselected frequency with a 5 Hz window (± 2.5 Hz). Therefore, it is possible that alpha power (~10 Hz) contaminated recordings at 13 Hz. However, our results showed increased power at night even in frequencies above 14 Hz, which are unlikely to be contaminated by alpha power (Fig. 2A).

In our study, contrary to most studies on sleep in PD, neural activity was recorded chronically while the subjects were on typical medication dosages and on-stimulation. Thus, to further understand the variability in circadian rhythms, especially in the beta band, we investigated the effects of both medication and stimulation on circadian beta power. We found that the nighttime use of extended-release

levodopa modulated the direction of power changes in the GPi (increase vs. decrease at nighttime). In particular, a reduction in beta power at night was more likely observed in subjects treated with extended-release levodopa when compared to those who were not taking extended-release levodopa. The use of extended-release levodopa could suppress beta power at nighttime similar to the daytime[27–29], thus masking the re-emergence of pathological beta power at night as seen in subjects who were not taking extended-release levodopa. This is supported by the baseline recording comparison, showing that subjects with increased power at nighttime had significantly higher power in the awake off-medication off-stimulation state.

Another important consideration is the potential contributions of sleep spindles, a typical electrophysiology feature (10–15 Hz) present in electroencephalograms (EEG) during sleep, specifically the NREM phase[30,31]. Sleep spindles are considered instrumental in memory consolidation, and motor learning increases sleep spindle density[32]. In addition to the EEG studies, deep brain recordings have also captured sleep spindles in both non-human primates[33] and humans[34]. Although sleep spindles have been primarily studied in the thalamus, given the thalamic–basal ganglia connection, it is possible that sleep spindles could affect chronic nighttime recordings in the GPi[35]. Previous studies have found that high frequency (≥130 Hz) DBS may have either an enhancing effect on sleep spindle density[36] or no significant effect on sleep spindle density between ON and OFF DBS states[37], but none reported a reduction of sleep spindles. In our study, chronic recordings in the low beta band likely overlapped with the sleep spindle spectral ranges. Although dopaminergic medication is known to reduce beta power, its effect on

sleep spindles is not well understood. Therefore, it is possible that sleep spindles might have contributed to the increase in power at night observed in some subjects. Future prospective studies are needed to fully address the presence of sleep spindles and their impact on chronic nighttime recordings.

Our observations of how circadian rhythms impact basal ganglia activity have important potential implications for the implementation of aDBS therapy in PD. Currently, aDBS algorithms use a single or double threshold approach that is centered around the pathological beta signal. Our study has revealed that spectral power in the basal ganglia, specifically in the beta band, showed different trends in different subjects. Additionally, our results suggest that extended-release levodopa may contribute to changes in circadian rhythms, meaning that aDBS algorithm may need recalibration for nighttime therapy when medications change. Thus, we caution that while aDBS can improve the responsiveness of neuromodulation therapy in PD, the algorithms will need to be tailored to each subject's neural signature for optimal therapeutic benefit and circadian patterns are an important consideration.

This study has inherent limitations associated with its retrospective design that need to be highlighted. First, we did not have wearable sleep data or formal sleep diaries to confirm the sleep/wake states in all subjects. Instead, we used a fixed time-of-day window for nighttime and daytime analysis, and all findings were framed as daytime/nighttime circadian rhythms, instead of sleep/awake cycle, similar to a previous chronic STN circadian rhythm study[12]. Another limitation is the potential artifacts from recording in unrestricted naturalistic conditions. We excluded recordings from the delta/theta ( < 8 Hz) band to avoid any artifacts from electrocardiogram (ECG) commonly seen in Percept devices with legacy Medtronic electrodes[38]. Upper body motion can also lead to broadband motion artifacts[39] which might inflate the sensing power measurements, especially during the awake period. In some cases, these artifacts could have caused an artificial power reduction at nighttime. Additionally, we did not have real-time measurements of the subjects' symptoms, disease states, or sleep stages at the time of recordings. Finally, the temporal resolution of these recordings was limited to a preselected band power averaged over 10 min that prevented us from studying fast changes in GPi LFP signals that might have occurred at night and/or during the day.

Future clinical studies will need to better characterize the relationship between raw GPi LFP signals and awake/asleep states, sleep stages, PD symptoms, and related parasomnias. These findings could be used to help guide the use of circadian rhythmicity in aDBS strategies.

We conclude that spectral power recorded chronically from the GPi in subjects with PD shows circadian rhythms, and the directionality and amplitude of the circadian rhythms depend on the frequency band and the use of extended-release levodopa. Circadian rhythms will need to be accounted for in aDBS algorithms to ensure successful implementation in naturalistic, at-home environments.

## Methods

### Study participants and data collection
This retrospective study was performed at the Norman Fixel Institute for Neurological Diseases at the University of Florida (UF). All subjects provided informed consent in accordance with the Declaration of Helsinki to participate in the UF INFORM Database, a database that stores subjects' clinical records and neural recordings[40]. This study was approved by the UF Institutional Review Board (IRB) to request and process data from the UF INFORM Database. The inclusion criteria were individuals diagnosed with PD by a movement disorders-trained neurologist and who were implanted with unilateral or bilateral electrodes in the GPi and attached to the Medtronic Percept PC. This sensing-enable neurostimulator[41] is capable of recording time–domain local field potentials (LFP) in-clinic (at a 250 Hz sampling rate) and neural power in a preselected frequency band (bandwidth ±2.5 Hz)

every 10 min for up to 60 days. Subjects included in this study had neural power recorded for at least 5 consecutive days. All subjects were programmed in a monopolar stimulation (stimulation contact 1 or 2), allowing sensing from electrodes surrounding the stimulation contact to reduce stimulation artifacts[42]. The spectral frequency band selected for chronic recordings usually corresponds to the most prominent peak observed in the spectral power of baseline surveys[41]. The baseline recordings were acquired in the clinic and in the off-medication and off-stimulation state, while chronic recordings were done in the subject's home environment with medication and DBS on. Neural recordings extracted from the UF INFORM Database were imported into the Brain Recording Analysis and Visualization Online (BRAVO) Platform[43]. Given the variability in the duration of the recordings (spanning a few days to several months) for each subject and each hemisphere, the first 5 consecutive days of chronic neural recording with stable stimulation and sensing configurations were selected. Among the 347 subjects with PD with the Percept neurostimulator and who have consented to be part of the UF INFORM Database, 93 subjects (130 unique hemispheres) met the inclusion criteria for this study.

In addition to neural recordings, the subject's demographics, therapeutic DBS parameters, and medications at the time of the recording, as well as the preoperative symptom assessments (UPDRS part 3) were extracted from the UF INFORM Database and via chart review. Therapeutic stimulation frequency, pulse width, and amplitude were used to calculate the TEED using a method previously described in Koss et al.[25]. For each subject, the dopaminergic and non-dopaminergic medications, dosage, and schedule were extracted, and the LEDD was calculated using the conversion factors estimated by Tomlinson and colleagues[22,23]. In addition, the time of medication was used to determine if subjects were taking extended-release levodopa and/or non-levodopa sleep medications (melatonin, benzodiazepines) at nighttime (defined as a nightly dose or scheduled at least 3 times a day). The preoperative UPDRS scores were used to determine the PD motor phenotype (tremor dominant, akinetic-rigid, intermediate) based on ref. 24. The stimulation location was defined as dorsal if contact 2 was used for stimulation and ventral if contact 1 was used.

Since this was a retrospective study, wearable devices and sleep diaries were not available. However, the Percept allows clinicians to set up event marking on the device, which allows subjects to mark specific events (e.g., 'medication intake, 'presence of dyskinesia') that might be relevant for clinical care. Therefore, the time of each event marked during each recording was extracted and used as a surrogate for wakefulness. Although every subject was instructed to mark events whenever they occurred when out-of-clinic, the event markings were typically sparse and greatly varied across subjects.

### Data processing
Data were preprocessed to handle potential outliers and missing data points. Outliers were defined as more than 5 standard deviations from the mean power recordings per subject. If outliers were detected, the values were removed and replaced by interpolating the previous and next data points. Recordings with missing data points were excluded. All spectral powers were z-score normalized using ±12 h of recordings to eliminate non-stationary slow-wave drifts and facilitate group analysis. All neural recordings were time-aligned on a 24-h clock based on Eastern Daylight-Savings Time (EDT, GMT−04:00). Nighttime power and daytime power were extracted by taking the average power between 0:00 am to 5:00 am and 3:00 pm to 8:00 pm, respectively. The circadian change in power was calculated as the mean normalized power during daytime subtracted by the mean normalized power during nighttime across all recorded days (day−night). A population-level circadian heatmap was generated to visualize the normalized 24-h power from all unique hemisphere recordings by grouping based on sensing frequency bands and sorting by the direction and magnitude of the effect of circadian rhythm to show the proportion of

subjects with an increase of power in nighttime versus subjects with a decrease of power in nighttime.

In a subset of subjects, neural time–domain signals were also recorded from each sensing-enabled bipolar contact, in-clinic and in the off-medication and off-stimulation condition while the subject was at rest. The power spectral density (PSD) was computed with a 1000 ms window and 500 ms overlap with the Hanning window. The PSD was then averaged over the beta band and compared between subjects who had increased power at nighttime and subjects who had decreased power at nighttime.

### Statistical methods
Shapiro's normality test was performed on the normalized daytime/nighttime power change to determine the appropriate statistical tests. The test statistics concluded that the data was normally distributed ($p > 0.05$); therefore, parametric tests were performed as described below.

First, we studied the relation between frequency band sensed and circadian rhythms. Each recording was categorized into alpha (7–12 Hz), low beta (13–20 Hz), and high beta (21–30 Hz) groups depending on the frequency band sensed. A one-way ANOVA was used to examine the effect of sensing frequency on circadian rhythms. Post-hoc pairwise comparisons between each group were performed using Tukey's test if a significant effect was found from the ANOVA. All *p*-values were corrected for multiple comparisons with Bonferroni correction.

Second, a GLM was built to examine the factors contributing to circadian rhythms within the beta band (13–30 Hz). The GLM was constructed with the normalized changes in the beta power during daytime versus nighttime as the dependent variable. Independent variables for the GLM included the therapeutic settings (dorsal versus ventral stimulation location), the TEED, LEDD[22,23], the use of nighttime extended-release levodopa, the use of non-levodopa sleep medications, the preoperative UPDRS part III scores, and the PD motor phenotype (tremor, intermediate, or akinetic subtype). The use of nighttime medication was defined as specific pharmaceutical instruction of medication taken nightly or the last dose of a periodic schedule that occurred more than three times per day.

Lastly, a non-parametric non-paired Mann–Whitney *U* test was used to compare the baseline beta power (recorded in clinic in the off-medication off-stimulation) between subjects showing increased beta power vs those with decreased beta power at nighttime.

### GPi lead localization
Preoperative T1 MRI images and postoperative CT images were extracted for each PD subject with sensing frequency in the beta band (13–30 Hz). Each subject's CT was coregistered to their preoperative T1 MRI using the rigid registration algorithm by Advanced Normalization Tools (ANTs)[44]. Then, lead locations were localized manually within the electrode artifact in the postoperative CT using 3D Slicer software[45], and the coordinates of the center of the bipolar sensing contacts was saved. The standard brain space (MNI152 Template[46]) with the corresponding DISTAL Atlas[47] for pallidal segmentations were then non-linearly transformed to the T1 MRI images. The transformation was applied to the sensing locations and presented in Fig. 3. Sensing locations with increased power at nighttime (blue) and decreased power at nighttime (red) did not show clear differences.

### Statistics and reproducibility
The goal of this retrospective study was to investigate circadian rhythms in the basal ganglia in subjects with PD. Therefore, no statistical methods were used to predetermine sample size, all data meeting the inclusion criteria were included, and no data were excluded from the analyses. The study was an observational retrospective study and not a clinical trial. Therefore, the study was not randomized or blinded, and the participants did not receive compensation for their participation.

### Reporting summary
Further information on research design is available in the Nature Portfolio Reporting Summary linked to this article.

## Data availability
The data generated in this study are provided in the Supplementary Information and Source Data file. The raw (identifiable) data are protected and are not available due to data privacy laws. The processed de-identified data will be shared upon request. Note that these datasets are part of ongoing research study. Any additional requests for information can be directed to, and will be fulfilled by, the corresponding authors. Source data are provided with this paper.

## Code availability
Analysis codes generated in this study are available at https://github.com/Fixel-Institute/Publication-Scripts. The BRAVO platform (Brain Recording Analysis and Visualization Online), used for neuronal data extraction and preprocessing, has been previously published and shared as an open source tool[43].

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

## Acknowledgements

The authors would like to thank the providers from the Norman Fixel Institute for their assistance with data collection. We also thank Chuck Jacobson and his team for managing the UF INFORM database. This project was not funded.

## Author contributions

J.N.C.: Conceptualization, Data collection and curation, Formal analysis, Data interpretation, Writing manuscript. T.d.A.: Data collection and curation. K.A.J.: Data interpretation, Editing manuscript, J.Y.: Clinical Feedback, Data collection, Editing manuscript. L.F.: Clinical Feedback, Data collection, Editing manuscript. F.S.: Data collection, Editing manuscript. J.K.W.: Clinical Feedback, Data interpretation, Editing manuscript. S.J.L.: Clinical Feedback, Data interpretation, Editing manuscript. M.S.O.: Editing manuscript. C.D.H.: Conceptualization, Data interpretation, supervision, Writing manuscript.

## Competing interests

The authors declare no competing interests.
