## [Peer Review File · Nature Communications]

Chronic intracranial recordings in the Globus Pallidus reveal circadian rhythms in Parkinson's diseaseREVIEWER COMMENTS

Reviewer #1 (Remarks to the Author):

Authors of “Chronic Intracranial recordings in the basal ganglia reveal patient specific circadian rhythms in Parkinson’s disease” conducted a retrospective analysis of 119 subjects with Parkinson’s disease (PD) treated with either subthalamic deep brain stimulation (STN-DBS; 23 subjects) or globus pallidus internal segment (GPI; 96 subjects) DBS. Notably, for this study all patients received DBS treatment through a stimulator battery that allowed for bidirectional stimulation and recording. The recording feature of the battery was used to collect two forms of electrophysiological data 1) chronic local field potential (LFP) data within a narrow frequency band (5 Hz range) across several days of in-home collection and 2) subject marked events. Currently, this is the largest dataset to date to explore these in-home LFP recordings from DBS subjects. From these two recording locations, authors observed three main findings 1) both GPI and STN activity were significantly altered by circadian rhythms and 2) the direction of the circadian rhythm depended on the frequency band for both brain regions, and 3) use of long-acting dopaminergic medications significantly impacted the circadian rhythms of the GPI target. This work provides important insights and considerations related to addressing the impact of circadian LFP features for developing adaptive stimulation protocols. Overall, they demonstrate that LFP power fluctuates across the day-night cycle, though the direction and magnitude of this fluctuation is more variable in the GPI than the STN. LFP power in different frequency bands also fluctuates differently, with high beta and gamma LFPs more likely to decrease at night in GPI and both low and high beta more likely to decrease at night in STN. To examine reasons for this variability, a linear model was constructed that shows that the use of extended-release levodopa is associated with a higher likelihood of reduced beta LFP power at night. There are many challenges to deriving useful information from observational data collected opportunistically from a highly heterogeneous population. The authors made an admirable effort in synthesizing these data and attempting to draw interpretable insights. However, the significant heterogeneity, limited consistency in comparisons, concerning circularity in their methodology for defining day vs night (or sleep vs awake) and largely generic conclusions limit the enthusiasm of the work and generalizability of its insights.

Major concerns:

1. The manuscript does not have a clear focus nor objective and in the absence of a clear hypothesis, the observations provide limited insight to the field. The Abstract and Introduction allude to using these data and results to inform adaptive DBS (aDBS), but it is not clear how authors linked their observations to strategies or challenges in aDBS, beyond stating that the noted variability will be problematic. A significant portion of the Introduction is allocated to highlighting the usefulness of aDBS; however, the results and figures do not appear to link back to aDBS. Further, the challenges inherent to interpreting retrospective observational data in this study appear to have made it difficult to draw a clear conclusion. Given the lack of control over data collection the authors are necessarily left with making vague generalizations about the data that fail to provide any novel insights over previously published work. For example, in the Abstract, authors conclude that “findings demonstrate the variability of chronic circadian rhythms and suggest that aDBS will likely need to account for these patient-specific fluctuations”. The reader is left with the conclusion that chronic recordings are variable and patient specific. Further, the

objective of the study as stated in the last paragraph of the introduction is “to characterize circadian patterns of neural activity recorded in the GPi and STN in a large cohort of subjects with PD...” This is neither goal-directed, nor hypothesis driven and fails to address a lack of knowledge in the field.

2. The lack of sufficient controls, conditions, and data representation significantly impacts the usefulness, generalizability of the observations and insights. For example, given the significant disparity between STN and GPi, should the authors have instead focused just on GPi. There is very little accounting for chronotype and the daily living activities of the subjects which will necessarily affect sleep data; this confound extends even to selection of data for analysis including the selection of number nights, and when those nights occurred. In the absence of any external metric (beyond event markers) for sleep behavior (see point #3) it is very difficult to interpret the findings in the study.

3. I have significant concerns regarding the assessment of day vs night as a proxy for sleep vs awake as described in the Methods. Admittedly, I may not be familiar with the methods as described, but I will attempt to delineate my concerns about how the methods impact interpretation of the results. Authors note in the results that the “directionality and amplitude of the changes [of circadian rhythms in GPi and STN of PD patients] depend on the frequency tracked” and that the “variability in circadian rhythm (increase vs decrease power at night)”, ultimately concluding that they both increase and decrease in circadian rhythm during the “night”. However, these results are predicated on defining day vs night (sleep vs awake) using the LFP power – “The time of ‘low’ and ‘high’ neuronal signal periods was used to determine when each period occurred (day vs night)”. The authors have no objective way to determine when in fact their subjects were awake or asleep. In some cases (61%), they have infrequent event markers, but this does not occur but a few times a day and not for all days, so at best, it would give a rough approximation of day vs night but not well-defined borders of time. The issue for the authors, as it appears to me, is that they are attempting to make inferences and draw conclusions about an observed phenomenon (i.e., changes in LFP regarding day vs night) by using the observation as the method of definition (i.e., increases and decreases in LFP define day vs night). This seems like a serious issue for the methodology and interpretation. Authors describe a process of timeline smoothing and normalization predicated on 2 hour and 30-minute bins, however, there is no justification for these parameters and the resolution of these data is low (10 minutes). So, a 30-minute bin would only have 3 data points. Further, there is no explicit description of how artifacts in the timeline data were managed; it is common to have outlier values and missing values in the 144 possible samples per day. Ultimately, the methodology used to define day vs. night based on LFP power could be misleading, as this approach assumes a direct correlation between LFP power and sleep-wake states without external validation. This assumption can lead to inaccurate interpretations, especially in cases where LFP patterns do not align with typical sleep-wake cycles.

4. There is a significant concern related to the interpretation of the medication impact on the reported results - Line 197: It would be helpful to further define how medications like extended-release levodopa and benzodiazepines were being used. For example, extended-release levodopa used multiple times per day may not have any significant effect when comparing beta frequency LFPs between day and night. But if the extended-release levodopa were used exclusively at bedtime, one might expect that this would influence the circadian cycle of LFPs. Furthermore, from the Discussion, Line 250: As above, the statement “The use of long-acting levodopa medications could suppress beta power at night-time” seems to imply that participants in this study were taking extended-release levodopa at night only, but is this known to be true? If so, it should be stated explicitly. If not, this is a limitation of the analysis discussed here and presented in Figure 4

5. Throughout the manuscript, authors conflate day/night with awake/asleep (e.g., lines 355-357). These classifications carry significant implications for the conclusions the authors attempt to draw from the LFP data.

Minor concerns:

Abstract:

1. This phrasing “Both GPi and STN activity was significantly altered by circadian rhythms” implies causality or at least some baseline comparison or manipulation.
2. The statement “depended on frequency band in both the GPi and STN,” should be clarified with at least low or high frequency bands or just noting the exact frequency band (i.e., alpha).
3. The “direction of circadian rhythm” should be clarified. I appreciate that there is parenthetical explanation, but the phrasing is ambiguous. I would suggest stating ‘Whether activity increased or decreased during the night...’.
4. Authors should consider including the number of individual hemispheres for STN vs GPi in abstract.

Introduction:

1. Authors should consider the relevance of the aDBS paragraph to the objectives, analysis and Figures/Results contained in the study. Given that the authors are retrospectively describing circadian patterns in these subjects, and not focused on subjects currently receiving aDBS therapy, it is possible that the theoretical impact of circadian cycling variability on aDBS are greater than would be expected if the study had enhanced control over LFP frequency selection, and subject selection (e.g., subjects may be excluded if they do not have a SenSight lead). It remains unknown how aDBS might be implemented, there are several possible strategies, some of which involve LFP, some of which may involve wearable devices that account for sleep/wake activity. The current Introduction appears to assume reliance on only LFP.
2. Line 75: The authors state that “A recent study evaluating chronic STN LFP recordings in the home environment has revealed beta power circadian fluctuations, with a consistent reduction of beta power during sleep” and cite the recent paper by van Rhee et al., npj Park Dis, 2022. However, it should be noted that this study showed that beta power was reduced during the overnight hours, but there was no external validation of sleep-wake state, so it cannot be stated with certainty that the reduction in beta power occurred during sleep.
3. Line 77: A reference (or multiple) should be provided for the statement “This result supported prior findings from acute STN recordings, performed in-clinic using externalized electrodes over 1 night, and showing that, on average, beta (13-30Hz) and gamma (30-60Hz) power in the STN decreased, while delta (0-3Hz), theta (3-7Hz), and alpha (7-13Hz) power increased during non-rapid-eye-movement (NREM) sleep when compared to wakefulness and rapid-eye-movement (REM) sleep”.

Results:

1. Why did the authors choose the described data collection window (i.e., 13 ± 35 days after pulse generator implantation)? Was the centering time point, the first programming session?
2. Line 117: It should be stated whether the frequency band for chronic recording was chosen in the on-medication state or off-medication state, as doing so in the on-medication state might suppress beta activity and thus skew the results, particularly those presented in Figure 2.
3. Authors note that “the bipolar contact pair used for sensing was determined by the contact(s) used for continuous stimulation therapy”. Was this an inclusion/exclusion criterion? In the window of time used for subject selection, how many subjects were excluded because they did not have sensing compatible

therapeutic configurations?

4. Authors should clearly describe the fractions for each frequency band bin across hemispheres within region. The band with the greatest fraction is noted for each band, but the remaining bands are not reported.

5. Line 122 – ‘reordered’ should be ‘recorded’

6. It appears that the number of subjects with gamma localized peaks was low and targeted based on documentation of dyskinetic peak which differs for peak determination in all other cases. Is it reasonable to include this group/frequency band in the analysis?

7. For the representative subjects highlighted in Figures 1 and 2, authors highlight 5 days. Authors should indicate in parentheses in the text and Figure legend the total number of days recorded for each subject and where in the context of the overall recording these 5 days were selected. Why 5 days, were they weekdays? Authors should note which hemisphere was recorded for each of the two representative Figures.

8. Line 170: It is stated that “a one-way ANOVA to compare across spectral bands revealed a significant effect of sensing frequency in GPI”. It is assumed that the comparison here was the effect of sensing frequency on the difference in LFP power between day and night, but this should be stated explicitly. The same is true for the next paragraph (line 181).

9. The use of non-parametric analyses for individual bands and a parametric analysis across bands seems arbitrary. Why did authors choose a non-parametric analysis? Is it reasonable to compare against 0? Why not day vs night? For the ANOVA, it is not clear how data were organized for the analysis. There are known asymmetries between frequency band and night, how did authors organize the data into discrete groups for the ANOVA?

10. Line 180: Provide the p-value for theta/alpha power.

11. Line 195: The linear model is described as determining the effect of several variables on the “change in beta power”. More clarification is needed on what exactly the dependent variable is here – is it the magnitude of the difference in beta power between day and night? Or the direction of change (greater during the day or greater at night)? Or something else?

Methods:

1. There is a discrepancy in the number of subjects reported between the Abstract and Introduction (n = 119) and the Methods (n = 117).

2. Missing demographic data: fraction of unilateral vs bilateral, fraction of SenSight vs legacy lead, bipolar vs monopolar stimulation configurations.

3. No clear description of which patient specific events were selected and how patients were advised to use those events

4. No description of how artifacts or missing data were handled in the Timeline data.

5. Line 315: A definition (local field potential power) and appropriate units (uVp) should be provided for “brain recording”. This applies also to line 318, lines 341-348, and lines 356-357 (“neural signal”).

6. Line 317: The sensing-available configurations should be described, as not all readers may be familiar with this.

7. Line 324: Either the method used to calculate the LEDD should be described, or a reference should be provided for the calculation.

Discussion:

1. Line 291: It is stated here that stimulation parameters (i.e., amplitude, pulse width, etc.) were not available in this dataset, but in the Methods it is reported that “therapeutic variables such as therapy

frequency, therapy pulse width, and therapy amplitude were extracted” (line 366). These statements seem contradictory and should be reconciled.

2.

Figures

1. For both Figures 1 and 2:

a. There’s no indication of what events were marked by the red-vertical lines – are they all the same, different event types

b. There is no scale marker on the polar plot radial axes

c. It is not clear from the methods how their day / night algorithm would successfully differentiate between a ‘sleep period’ that in one patient LFP went up vs one that LFP went down when their algorithm appears to assume high activity periods are day/awake related (see major comment #3).

d. There’s no description in the Results/Methods/Figure legend for how the plots in C were generated.

Does each row represent the average for all days recorded for one hemisphere? How did authors control differences in the number of days?

2. Figure 3 is labeled “Figure 2”. Additionally, would suggest using different symbols to denote different statistically significant findings and defining what each symbol means (eg, * is used twice in panel 3B, and ** is also used – what do these indicate?). Similarly, no explanation is given for the * symbol in panel 3D.

3. Figure 4 is labeled as “Figure 3”, and is referred to in the text as “Figure 3C”.

4. Table 2: Variables included in the linear model should be defined, particularly “channel” as this is not described in the text and it is unclear to what this is referring.

5. Supplementary Figure: The legend states that the circadian beta rhythm was strongly diminished by the increase in stimulation amplitude, but is it possible that this is just a floor effect? That is, the top panel clearly shows LFP power is overall reduced by the increase in stimulation amplitude, thus making the absolute difference between day and night smaller, but is the relative difference changed at all?

Visually, it appears that there is still some rhythmicity to the LFP power.

Grammar/Syntax:

Line 111: Clarify “out stimulation settings through the recordings.”

Line 122: Change “reordered” to recorded.

Line 183: Change “Figure 3E” to Figure 3D.

Line 238: Change “(±2Hz)” to (±2.5Hz).

Line 323: Change “proxi” to proxy.

Table 1: Change “LEED” to LEED.

Reviewer #2 (Remarks to the Author):

In this study, Cagle et al reported a large retrospective cohort including 119 subjects with PD (165 hemispheres) with GPi and STN activity recorded chronically in the home environment. They find that both GPi and STN power are significantly modulated by circadian rhythm. Increased power was also found, especially in GPi (28%) at night, and both the frequency of the power band and the use of long-acting levodopa medications influenced the neurophysiological circadian rhythm. Overall, this is a well-powered study with the largest known subject numbers with chronic in-home recordings. However,

some areas warrant constructive criticism and further consideration.

First, analyzing diurnal fluctuations associated basal ganglia oscillations has been conducted in previous papers^{1,2}. Sleep-related modulation in the pallidum has also been looked at in two recent studies^{3,4}. I take the major novelty of the work presented by the author as the large sample size included, and the finding that beta power during sleep could be higher than that during wakefulness (in 1/3 of the GPi recordings), an observation that has never been reported before. However, regarding this finding, a) the authors did not provide sufficient explanations for this phenomenon in the discussion, e.g., why beta power could be even higher during sleep than during wakefulness; what trait do these 1/3 patients have that differentiates them from the others? And b) the authors did not provide in-depth data supporting this finding and investigating what caused the beta enhancement, e.g., would it be in any case the beta elevation is caused by any type of artifact? Does beta burst have a longer duration during sleep for the beta-elevated group? During which sleep stage does the beta enhancement occur? Without more detailed information, it's hard to judge the validity of the results and the implication of the observed beta enhancement at sleep.

I personally would suggest the authors compare the clinical and oscillatory features between the beta-increased and the beta-decreased groups to see what caused the difference. I would also suggest the authors record raw LFP data from at least several of these beta-increased patients to investigate how beta is changed during sleep-awake transitions. To the best of my knowledge, all published studies hold that beta power decreases from wakefulness to NREM sleep, and increases prior to awakening. If the presented results by the authors are valid, one possibility for the increase of beta during sleep could potentially be caused by the sleep spindle (11-15 Hz) and its higher-order harmonics^{5,6}. As it has been shown that sleep spindle exists in basal ganglia and could have an impact on the basal ganglia electrophysiological patterns. Another possibility could be that beta increased dramatically during REM sleep at night and in these beta-increase patients the proportion of REM sleep was exceptionally long. While in the daytime, beta was suppressed by medication and stimulation, resulting in the observation that beta increased at night. However, with only an average power every 10 min, it's hard to test these hypotheses.

Second, the heat maps shown in Figure 1 and Figure 2 are somewhat misleading. If I understand correctly, the y-axis is the power of the recorded band but not necessarily the beta power, despite that the authors present examples of beta increase and decrease on the left panels. In fact, as the authors reported, in the GPi group, in 46.7% of the hemisphere the recorded frequency band is the theta/alpha band. It could be pointless to plot different frequency bands together and combinedly term them "recordings", especially when the direction of modulation could oftentimes be opposite for different bands. I suggest plotting separate figures for each individual band.

Third, the automated circadian detection algorithm developed by the authors seems to be insecure to me. The authors used the time of 'low' and 'high' neuronal signal periods together with the event marked to determine day vs night. What if the patients had insomnia and could not get to sleep at night (strong beta power observed), but had too much daytime sleep (weak beta power observed) with the events representing only the time points that they were awakened to take medicine? Indeed, the beta power trace shown in Figure 2a could probably be explained by this. In this scenario, the automated detection algorithm gives erroneous labels.

Fourth, since the subthalamic and pallidal signals are influenced by the extended-release levodopa, the authors need to give more details about how they take this into consideration into their sleep/wake

classification algorithm, e.g., showing the time of extended-release levodopa taken and how they influence the 24-hour spectrogram and the corresponding classification results.

Other points:

1. Did the authors collect sleep related scales, e.g., PSQI, PDSS, RBDSQ, for the patients? Which could help explain the differential modulation of beta power between subjects.
2. The number of GPi implantations is much larger than the number of STN implantations. The authors should report how they choose between the two targets. Would this cause differences in clinical features between groups?
3. The bipolar contact pair used for sensing was determined by the contact(s) used for continuous stimulation therapy. Could the author show the contact location of these patients, especially comparing the location between the beta increased and decrease patients.
4. In supplementary figure 1, I noticed that an increase in stimulation amplitude is also associated with a slight but consistent increase in the beta power during sleep. Given the y-axis is absolute power rather than z score power, how do the authors explain this beta increase after turning up the stimulation amplitude?
5. In most sections, 119 subjects were mentioned, while in the Methods section it was 117 subjects. This discrepancy should be checked.
6. There are two Figure 2 in the manuscript. The latter one should be Figure 3.

References

1. Baumgartner, A., Hirt, L., Kern, D. & Thompson, J. Diurnal fluctuations of local field potentials follow sleep-wake behavior in Parkinson's disease. (2023). doi:10.21203/rs.3.rs-2468375/v1.
2. van Rheede, J. J. et al. Diurnal modulation of subthalamic beta oscillatory power in Parkinson's disease patients during deep brain stimulation. *NPJ Parkinsons Dis* 8, 88 (2022).
3. Yin, Z. et al. Pallidal activities during sleep and sleep decoding in dystonia, Huntington's, and Parkinson's disease. *Neurobiology of Disease* 182, 106143 (2023).
4. Yin, Z. et al. Pathological pallidal beta activity in Parkinson's disease is sustained during sleep and associated with sleep disturbance. *Nat Commun* 14, 5434 (2023).
5. Zhou, Y. et al. Methodological Considerations on the Use of Different Spectral Decomposition Algorithms to Study Hippocampal Rhythms. *eNeuro* 6, ENEURO.0142-19.2019 (2019).
6. Sushkova, O. S., Morozov, A. A. & Gabova, A. V. EEG Beta Wave Trains are not the Second Harmonic of Mu Wave Trains in Parkinson's Disease patients. in (2017).

Reviewer #3 (Remarks to the Author):

This study, by Cagle and Colleagues, comprises retrospective analysis of a very large data set recorded from people with Parkinson's disease that have been implanted with the Medtronic Percept device. The power of such recordings is that they allow spectral power at preset frequency bands to be collected at

regular intervals for weeks and months, giving unique insight as to how these activities are modulated by circadian and diurnal rhythms. Studies using these data are still relatively rare and few have utilized the impressive numbers of patients reported here. In addition, the investigators report recordings from both STN and GPi. They also have self-reported measures of wakefulness, adding a measure of ground truth that has been absent from most other studies that have used these methods. The authors demonstrate that, as previously reported, in many patients “beta” power decreases during sleep. However, they show that this relationship is inverted with many patients and that the direction of the patients circadian modulation is dependent on the frequency of the beta power. This main observation is of potential importance to the development of closed-loop DBS approaches. However, there are currently several important caveats to the conclusions reached by the authors that considerably dilute its impact. Some of these issues are intrinsic to the methods used, for example that Percept data only allows one frequency band at a time to be analysed, while others could be addressed using further analysis.

Major Comments.

1. Are changes in diurnal beta profile driven by day time / waking differences or by night-time / sleep differences? The authors currently note two different diurnal profiles of beta, with their novel observation being that some show a relative increase during the night vs. the day and some show the opposite pattern. It is currently not addressed whether this is because of a day-time or night-time difference. A comparison of e.g. absolute power in comparable data sets (similar sensing frequencies) would be useful, to attempt to establish whether or not the novel finding represents increased beta during the night in some patients, or decreased beta during the day (or a mixture).

2. How can the authors discriminate between beta oscillations and sleep spindles. In the data sets from GPi and STN, it is very rare (Figure 3A&C) to see a night-time increase unless the sensing frequency is below 18Hz. As the sensing frequency band is 2.5Hz on either side of the selected frequency, this is exactly where the sensed ‘beta’ frequency starts overlapping with the frequency of sleep spindles (9-16Hz). Sleep spindles can lead to large spectral peaks during sleep and do propagate through the basal ganglia (e.g. DOI: 10.1016/j.celrep.2022.111367) so it is highly likely that they are driving at least some of the activity observed in this frequency band. While the authors do not have polysomnographic data and this is a retrospective study, some work could be done to address this. One approach would be to revisit the analysis of extended release levodopa; sleep spindles would not be expected to be suppressed by L-DOPA to the same extent as Parkinsonian beta. Another would be to provide more granular analysis of the night-time profile of patients with higher night time beta vs lower night-time beta – In the example in figure 2, high beta in STN appears to have more of a ‘peak’ and less of a ‘plateau’; is this representative of other STN or GPi data sets with high beta? This could be driven by sleep architecture (with spindles more common during some sleep stages, which are more common during certain parts of the night). Finally, addressing whether the changes in diurnal profile are driven by day time or night time ‘beta’ would also address this concern (similar absolute beta during the day but higher during the night would be expected if the effect is spindle-driven).

3. The clinical phenotype of Parkinson’s disease has a significant effect on the overall power of beta

oscillations in the LFP. Are the patients with inverse circadian profiles more likely to be tremor-dominant, which is associated with less beta? This could suggest that the night-time beta is more likely to be sleep-spindle related and may help to address the questions in point 2.

4. A discussion of data quality / potential artifact sources is also warranted given known risk of electrocardiographic and motion artifacts and the uncontrolled at-home nature of the recordings.

Minor points.

1. Are profiles patient-specific? Where there is data from 2 hemispheres of a patient, do the hemispheres show a correlation in diurnal profile? Does this depend on the similarity/difference in sensing frequency between hemispheres? If a high-beta peak in a patient shows low beta during the night, can a low beta peak in that patient's other hemisphere show an opposite circadian profile?

2. Introduction / discussion of canonical frequency bands. More effort should be made to discuss the meaning of activity in canonical frequency bands in the basal ganglia. The most obvious omission is the absence of a discussion of sleep spindles, which are an important potential confound. Other known changes in relevant basal ganglia activity bands could use some further discussion as well (e.g. differences between NREM/REM sleep stages, role during wakefulness, etc).

3. Figures 1C and 2C. Smoothing across patients (vertical axis) in figures 1C and 2C is not appropriate as it blends together independent patient data sets. Also, it would be much more informative to split these sorted profiles by canonical frequency band.

4. Supplementary figure 1

There is a clear reduction in circadian profile when the stimulation is turned up. It is interesting to see that it is not only driven by daytime beta being reduced, but also by night-time beta being increased. Can the authors comment on potential explanations?

How much data is available on the effect of stimulation

5. Methods - 'Day' vs 'night' period

Terminology is confusing; in figures 1 and 2 the authors call these periods 'sleep' and 'wake'. It is also not specified when the authors consider 'high neural signal' to mean day/wake or night/sleep – presumably this is dependent on the direction of the circadian profile? What is done for data sets where there were no significant differences between daytime and night-time? It should also be made a bit more explicit which analyses exactly these day/night or sleep/wake definitions are used for.

6. The figure font sizes are often very small.

7. This manuscript has many typos and needs a careful proofreading

Reviewer #4 (Remarks to the Author):

I co-reviewed this manuscript with one of the reviewers who provided the listed reports as part of the Nature Communications initiative to facilitate training in peer review and appropriate recognition for co-reviewers.

REVIEWER COMMENTS

Reviewer #1 (Remarks to the Author):

Authors of “Chronic Intracranial recordings in the basal ganglia reveal patient specific circadian rhythms in Parkinson’s disease” conducted a retrospective analysis of 119 subjects with Parkinson’s disease (PD) treated with either subthalamic deep brain stimulation (STN-DBS; 23 subjects) or globus pallidus internal segment (GPi; 96 subjects) DBS. Notably, for this study all patients received DBS treatment through a stimulator battery that allowed for bidirectional stimulation and recording. The recording feature of the battery was used to collect two forms of electrophysiological data 1) chronic local field potential (LFP) data within a narrow frequency band (5 Hz range) across several days of in-home collection and 2) subject marked events. Currently, this is the largest dataset to date to explore these in-home LFP recordings from DBS subjects. From these two recording locations, authors observed three main findings 1) both GPi and STN activity were significantly altered by circadian rhythms and 2) the direction of the circadian rhythm depended on the frequency band for both brain regions, and 3) use of long-acting dopaminergic medications significantly impacted the circadian rhythms of the GPi target. This work provides important insights and considerations related to addressing the impact of circadian LFP features for developing adaptive stimulation protocols.

We thank the reviewer for this summary of important findings from our study.

Overall, they demonstrate that LFP power fluctuates across the day-night cycle, though the direction and magnitude of this fluctuation is more variable in the GPi than the STN. LFP power in different frequency bands also fluctuates differently, with high beta and gamma LFPs more likely to decrease at night in GPi and both low and high beta more likely to decrease at night in STN. To examine reasons for this variability, a linear model was constructed that shows that the use of extended-release levodopa is associated with a higher likelihood of reduced beta LFP power at night. There are many challenges to deriving useful information from observational data collected opportunistically from a highly heterogeneous population. The authors made an admirable effort in synthesizing these data and attempting to draw interpretable insights. However, the significant heterogeneity, limited consistency in comparisons, concerning circularity in their methodology for defining day vs night (or sleep vs awake) and largely generic conclusions limit the enthusiasm of the work and generalizability of its insights.

Major concerns:

1. The manuscript does not have a clear focus nor objective and in the absence of a clear hypothesis, the observations provide limited insight to the field. The Abstract and Introduction allude to using these data and results to inform adaptive DBS (aDBS), but it is not clear how authors linked their observations to strategies or challenges in aDBS, beyond stating that the noted variability will be problematic. A significant portion of the Introduction is allocated to highlighting the usefulness of aDBS; however, the results and figures do not appear to link back to aDBS. Further, the challenges inherent to interpreting retrospective observational data in this study appear to have made it difficult to draw a clear conclusion. Given the lack of control over data collection the authors are necessarily left with making vague generalizations about the data that fail to provide any novel insights over previously published work. For example, in the Abstract, authors conclude that “findings demonstrate the variability of chronic circadian rhythms and suggest that aDBS will likely need to account for these patient-specific fluctuations”. The reader is left with the conclusion that chronic recordings are variable and patient specific. Further, the objective of the study as stated in the last paragraph of the introduction is “to characterize circadian patterns of neural activity recorded in the GPi and STN in a large cohort of subjects with PD...” This is neither goal-directed, nor hypothesis driven and fails to address a lack of knowledge in the field.

Response:

We thank the reviewers for helping us improve this manuscript with their constructive comments. We have modified the introduction to better describe the goal and motivation of this study. Most studies investigating the effect of circadian rhythms on LFP have been focused on STN activity, utilizing externalized electrodes. Recently, the chronic embedded neurostimulator has been used to investigate STN activity in the home environment. The physiology of GPi is still mainly unknown and all studies so far have investigated GP circadian rhythm in a controlled 'lab' environment and in the 'off medication' state. The goal of this study was to investigate the effect of circadian rhythms on GPi activity using chronic neural recordings in the home environment under naturalistic conditions. We here demonstrate that the STN and GPi have notably dissociated circadian patterns.

The introduction was modified, and clarification of the objectives of the study have been added: "The objective of this study was to characterize circadian patterns of neural activity recorded in the GPi using longitudinal, at-home recordings under naturalistic conditions (including unrestricted therapeutic medications and chronic stimulation conditions) via a sensing-enabled chronically implanted neurostimulator in a large retrospective cohort of subjects with PD." The abstract was also modified to clarify the goals: "Our objective was to characterize basal ganglia circadian rhythms in the GPi in a large cohort of Parkinson's disease (PD) patients in a naturalistic environment under therapy conditions."

2. The lack of sufficient controls, conditions, and data representation significantly impacts the usefulness, generalizability of the observations and insights. For example, given the significant disparity between STN and GPi, should the authors have instead focused just on GPi. There is very little accounting for chronotype and the daily living activities of the subjects which will necessarily affect sleep data; this confound extends even to selection of data for analysis including the selection of number nights, and when those nights occurred. In the absence of any external metric (beyond event markers) for sleep behavior (see point #3) it is very difficult to interpret the findings in the study.

Response:

We agree that the primary finding of this study relates to GPi data. Therefore, we are now focusing this manuscript on GPi physiology, presenting GPi-related results in the updated manuscript, while presenting STN results in the supplementary document, for control and comparison and validation of previous studies in the largest cohort to date, using an embedded, sensing-enabled pacemaker (in the stimulation ON and medication ON states).

To address the concern regarding data selection bias - for consistency, we now select the first 5 consecutive nights per hemisphere recorded without changes in stimulation settings or sensing settings. This did not significantly change the overall results of the study.

As correctly pointed by the reviewer one of the limitations of this retrospective study is the lack of objective measures for sleep vs activity. We acknowledge that there are substantial limitations to retrospective studies to investigate any behavior (including sleep), which we have outlined in full transparency in the 'limitations' section. Our goal with this retrospective study was to leverage a large cohort of patients to generate preliminary hypotheses about GPi circadian activity and how it could inform future iterations of DBS. In the future, we plan to pursue a prospective study using wearables, which will enable more objective and detailed sleep/awake characterization but will only be feasible to implement in a significantly smaller cohort.

This limitation section was modified: "It is important to highlight several limitations of our study. As a retrospective study, we did not have wearable sleep data or formal sleep diaries to confirm the sleep/wake states in all subjects and instead, we used a fixed time-of-day window for nighttime and daytime analysis and all findings were framed in the daytime/nighttime circadian rhythm, instead of sleep/awake cycle,

similar to previous chronic STN circadian rhythm study¹⁴. Another limitation is the potential artifacts from recording in unrestricted naturalistic conditions. Cardiac artifact is commonly seen in Percept devices with legacy Medtronic electrodes³⁸, and we excluded sensing recordings from the delta/theta (<8Hz) band to avoid contamination from electrocardiogram (ECG). Upper body motion can also lead to broadband motion artifacts³⁹ which might inflate the sensing power bands especially during the awake period. These artifacts may cause the recordings to display artificial power reduction at nighttime. Additionally, we did not have real-time measurements of the subjects' symptoms, disease states, or sleep stages at the time of recordings. Future clinical studies will need to better characterize the awake/asleep states, sleep stages, PD symptoms, and related parasomnias.”

regarding the assessment of day vs night as a proxy for sleep vs awake as described in the Methods. Admittedly, I may not be familiar with the methods as described, but I will attempt to delineate my concerns about how the methods impact interpretation of the results. Authors note in the results that the “directionality and amplitude of the changes [of circadian rhythms in GPi and STN of PD patients] depend on the frequency tracked” and that the “variability in circadian rhythm (increase vs decrease power at night)”, ultimately concluding that they both increase and decrease in circadian rhythm during the “night”. However, these results are predicated on defining day vs night (sleep vs awake) using the LFP power – “The time of ‘low’ and ‘high’ neuronal signal periods was used to determine when each period occurred (day vs night)”. The authors have no objective way to determine when in fact their subjects were awake or asleep. In some cases (61%), they have infrequent event markers, but this does not occur but a few times a day and not for all days, so at best, it would give a rough approximation of day vs night but not well-defined borders of time. The issue for the authors, as it appears to me, is that they are attempting to make inferences and draw conclusions about an observed phenomenon (i.e., changes in LFP regarding day vs night) by using the observation as the method of definition (i.e., increases and decreases in LFP define day vs night). This seems like a serious issue for the methodology and interpretation. Authors describe a process of timeline smoothing and normalization predicated on 2 hour and 30-minute bins, however, there is no justification for these parameters and the resolution of these data is low (10 minutes). So, a 30-minute bin would only have 3 data points. Further, there is no explicit description of how artifacts in the timeline data were managed; it is common to have outlier values and missing values in the 144 possible samples per day. Ultimately, the methodology used to define day vs. night based on LFP power could be misleading, as this approach assumes a direct correlation between LFP power and sleep-wake states without external validation. This assumption can lead to inaccurate interpretations, especially in cases where LFP patterns do not align with typical sleep-wake cycles.

Response:

Thank you so much for the comment on our method of defining daytime/nighttime periods. We understand the reviewer's concerns regarding our data-driven method which utilizes the differentiation of neural signals as the classifier and is a form of circular proof. We have updated our method to now use: 1) fixed time windows for nighttime (12am to 5am) and daytime (3pm-8pm), selected based on the assumption that patients will likely be asleep from 12am-5am and awake from 3pm-8pm; 2) using the raw signal (10-minute-averaged LFP power recorded on Percept PC neurostimulator) without any moving average low-pass filter. The results obtained with this new method are similar to those described in the original manuscript: the majority of GPi recordings showed a reduction of power at night while ~30% of the recordings showed an increased power at night. Similar observations were found with shorter windows from 12am-3am and 3pm-6pm, indicating that the results are robust to window size. Therefore, the manuscript has been modified to include this new method, and all figures have been updated. Additionally, the results emphasize differences in GP power at night vs day (instead of asleep vs awake).

Although missing data and outliers are common in time domain data collected with the Percept PC device, they are infrequently observed in frequency domain data collected chronically. The chronic power

recorded with Percept is an average of 10 minutes or 3000 sample points and is likely less susceptible to instantaneous artifacts and package loss than recording the instantaneous sample in the time domain. The pre-processing pipeline applied on this dataset includes removal of any outliers, defined as transient change with more than 5 standard deviations away from means, and linear interpolation of missing datapoints using previous and next datapoint. However, there were no outliers and no missing data in our dataset and therefore this method was not described in the initial manuscript.

To clarify, we have updated the manuscript to include a section describing the signal pre-processing.

“Data were pre-processed to handle potential outliers and missing data points. Outliers were defined as more than 5 standard deviations from the mean power recordings per subject. If outliers were detected, the values were removed and replaced by interpolating the previous and next data points. Recordings with missing data points were excluded.”

4. There is a significant concern related to the interpretation of the medication impact on the reported results - Line 197: It would be helpful to further define how medications like extended-release levodopa and benzodiazepines were being used. For example, extended-release levodopa used multiple times per day may not have any significant effect when comparing beta frequency LFPs between day and night. But if the extended-release levodopa were used exclusively at bedtime, one might expect that this would influence the circadian cycle of LFPs. Furthermore, from the Discussion, Line 250: As above, the statement “The use of long-acting levodopa medications could suppress beta power at night-time” seems to imply that participants in this study were taking extended-release levodopa at night only, but is this known to be true? If so, it should be stated explicitly. If not, this is a limitation of the analysis discussed here and presented in Figure 4

Response:

Thank you so much for the comment. We have now updated the manuscript to provide more details regarding patients’ medication states in both the ‘Results’ and ‘Discussion’ sections.

In summary, in this study, patients were either taking levodopa extended release (Levodopa ER) or levodopa immediate release (Levodopa) or both. Levodopa ER was either taken on a periodic schedule (at least 3 times a day), with the last dosage right before sleep or once before bedtime as instructed. This is now clarified in the manuscript.

This was clarified in the results section: “In particular, we investigated whether the change in beta power (dependent variable) between daytime and nighttime was significantly modulated by the following independent variables: the levodopa equivalent daily dose (LEDD)^{22,23}, the use of levodopa extended-release medication at nighttime, the use of non-dopaminergic medication at nighttime, the location of contact used for stimulation (ventral vs dorsal), the total electrical energy delivered (TEED), the Unified Parkinson's disease rating scale (UPDRS) total scores, and patient’s motor phenotype (tremor, intermediate, or akinetic subtype).” We also updated the discussion to provide more details on our interpretation.

5. Throughout the manuscript, authors conflate day/night with awake/asleep (e.g., lines 355-357). These classifications carry significant implications for the conclusions the authors attempt to draw from the LFP data.

Response:

We agree with the reviewer’s comment and have updated the manuscript and figures to describe differences in power in daytime vs nighttime (instead of awake vs asleep).

Minor concerns:

Abstract:

1. This phrasing “Both GPi and STN activity was significantly altered by circadian rhythms” implies causality or at least some baseline comparison or manipulation.

We have revised this sentence to: “Significant changes in GPi neural power were found between daytime and nighttime in the majority of the patients (82.3%).”

2. The statement “depended on frequency band in both the GPi and STN,” should be clarified with at least low or high frequency bands or just noting the exact frequency band (i.e., alpha).

We have revised this sentence to: “A reduction in power was observed at nighttime in 56.2%. However, power was increased at night in 26.2% of recordings, especially in the alpha and low-beta band.”

3. The “direction of circadian rhythm” should be clarified. I appreciate that there is parenthetical explanation, but the phrasing is ambiguous. I would suggest stating ‘Whether activity increased or decreased during the night...’.

To simplify the abstract, the revised abstract removed this phrase. The changes in circadian rhythm is described as: “A reduction in power was observed at nighttime in 56.2% of the patients; however, an increase in power was observed at nighttime in 26.2% of recordings, especially in the alpha and low-beta band.”

4. Authors should consider including the number of individual hemispheres for STN vs GPi in abstract.

We have included the number of GPi hemispheres in the abstract and put STN data in supplementary data: “This retrospective study includes GPi activity collected from 93 subjects with PD (130 hemispheres) and recorded chronically in the home environment.”

Introduction:

1. Authors should consider the relevance of the aDBS paragraph to the objectives, analysis and Figures/Results contained in the study. Given that the authors are retrospectively describing circadian patterns in these subjects, and not focused on subjects currently receiving aDBS therapy, it is possible that the theoretical impact of circadian cycling variability on aDBS are greater than would be expected if the study had enhanced control over LFP frequency selection, and subject selection (e.g., subjects may be excluded if they do not have a SenSight lead). It remains unknown how aDBS might be implemented, there are several possible strategies, some of which involve LFP, some of which may involve wearable devices that account for sleep/wake activity. The current Introduction appears to assume reliance on only LFP.

Thanks for all the suggestions. We have modified the introduction and have highlighted the goal and motivation of this study according to the reviewers’ suggestions: “The objective of this study was to characterize circadian patterns of neural activity recorded in the GPi using longitudinal, at-home recordings under naturalistic conditions (including unrestricted therapeutic medications and chronic stimulation conditions) via a sensing-enabled chronically implanted neurostimulator in a large retrospective cohort of subjects with PD.”

2. Line 75: The authors state that “A recent study evaluating chronic STN LFP recordings in the home environment has revealed beta power circadian fluctuations, with a consistent reduction of beta power during sleep” and cite the recent paper by van Rhee et al., npj Park Dis, 2022. However, it should be noted that this study showed that beta power was reduced during the overnight hours, but there was no

external validation of sleep-wake state, so it cannot be stated with certainty that the reduction in beta power occurred during sleep.

We agree that the study by Dr. van Rheede also did not provide external validation of sleep using wearables, similar to our study. We have modified this sentence to:

“A recent study evaluating chronic STN LFP recordings in the home environment has revealed beta power circadian fluctuations, with a consistent reduction of beta power during the overnight hours¹²” and have emphasize this challenge in the ‘limitations’ section.

3. Line 77: A reference (or multiple) should be provided for the statement “This result supported prior findings from acute STN recordings, performed in-clinic using externalized electrodes over 1 night, and showing that, on average, beta (13-30Hz) and gamma (30-60Hz) power in the STN decreased, while delta (0-3Hz), theta (3-7Hz), and alpha (7-13Hz) power increased during non-rapid-eye-movement (NREM) sleep when compared to wakefulness and rapid-eye-movement (REM) sleep”.

The references have been added.

Results:

1. Why did the authors choose the described data collection window (i.e., 13 ± 35 days after pulse generator implantation)? Was the centering time point, the first programming session?

At our institution, although clinicians often use chronic brain recordings, there is currently no consensus or specific protocol regarding the timing and duration of these recordings. In addition, typically, patients are asked to try different stimulation ‘group’ and have the ability to change the amplitude of the stimulation. Therefore, some patients might have only a few days of data recorded (with unchanged stimulation/recording settings), while others might have several months of data. Therefore, we have decided to select the first recording with 5 days of continuous recording without stimulation parameter changes and set recording settings. These 5 days of recordings occurred right after initial battery implantation (which happened 1 month after lead implant) and up to months after implanted. On average, the data were collected 59.9 ± 129 days after pulse generator implantation.

In our result and method section, we added that “To ensure data consistency, we selected the first 5 consecutive days (120 hours) of chronic sensing without changes in recording and stimulation settings for each subject and hemisphere. On average, the recordings occurred 59.9 ± 129 (mean \pm std) days after pulse generator device implantation, which is 4 weeks after lead implantation surgery.”

2. Line 117: It should be stated whether the frequency band for chronic recording was chosen in the on-medication state or off-medication state, as doing so in the on-medication state might suppress beta activity and thus skew the results, particularly those presented in Figure 2.

The frequency band was selected in the off-medication and is now clarified in the manuscript. “The frequency of oscillatory activity sensed chronically was selected in the off-medication state by the clinicians during a DBS programming session.”

3. Authors note that “the bipolar contact pair used for sensing was determined by the contact(s) used for continuous stimulation therapy”. Was this an inclusion/exclusion criterion? In the window of time used for subject selection, how many subjects were excluded because they did not have sensing compatible therapeutic configurations?

The goal of this retrospective study was to study GPi circadian rhythm. Therefore, only patients with LFP recordings were selected (and therefore only patients with sensing compatible configurations). We have clarified that in the method section. Initial query from our database returned 347 PD patients with the Percept neurostimulator at University of Florida, but only the final 93 patients in the manuscript contain sufficient chronic recording data.

“The inclusion criteria were individuals diagnosed with PD by a movement disorders-trained neurologist, and who are implanted with unilateral or bilateral electrodes placed in the GPi and attached to the Medtronic Percept PC... Given the variability in the duration of the recordings (from a few days to several months), for each subject and each hemisphere, the first 5 consecutive days of chronic neural recording with stable stimulation and sensing were selected. Among the 347 PD patients with the Percept neurostimulator and who have consented to be part of the UF INFORM Database, 93 subjects (130 unique hemispheres with recordings) meet the inclusion criteria for this study.”

4. Authors should clearly describe the fractions for each frequency band bin across hemispheres within region. The band with the greatest fraction is noted for each band, but the remaining bands are not reported.

We have added these numbers.

5. Line 122 – ‘reordered’ should be ‘recorded’

We have corrected this.

6. It appears that the number of subjects with gamma localized peaks was low and targeted based on documentation of dyskinetic peak which differs for peak determination in all other cases. Is it reasonable to include this group/frequency band in the analysis?

We agree with the reviewer that the number of gamma patients is low (6.1% of all recorded hemisphere (n=8)). For patients with sensing at gamma, most of them had additional chronic recordings that tracked non-gamma band(s). Therefore, we have updated our patient’s selection to focus on alpha (8-12), low beta (12-20), and high beta (20-30) band, which are commonly associated with PD bradykinesia and tremor electrophysiology. The manuscript methods, analysis and results have been updated.

7. For the representative subjects highlighted in Figures 1 and 2, authors highlight 5 days. Authors should indicate in parentheses in the text and Figure legend the total number of days recorded for each subject and where in the context of the overall recording these 5 days were selected. Why 5 days, were they weekdays? Authors should note which hemisphere was recorded for each of the two representative Figures.

Thanks for the suggestions. We have updated the figures and captions to provide more details on the example.

8. Line 170: It is stated that “a one-way ANOVA to compare across spectral bands revealed a significant effect of sensing frequency in GPi”. It is assumed that the comparison here was the effect of sensing frequency on the difference in LFP power between day and night, but this should be stated explicitly. The same is true for the next paragraph (line 181).

We have updated the manuscript accordingly: “A one-way analysis of variance (ANOVA) to compare across frequency bands revealed a significant effect of sensing frequency in GPi on daytime/nighttime power change ($F=18.22$, $p<0.001$).”

9. The use of non-parametric analyses for individual bands and a parametric analysis across bands seems arbitrary. Why did authors choose a non-parametric analysis? Is it reasonable to compare against 0? Why not day vs night? For the ANOVA, it is not clear how data were organized for the analysis. There are known asymmetries between frequency band and night, how did authors organize the data into discrete groups for the ANOVA?

We agree that the inconsistency of description for parametric and non-parametric test in the frequency band analysis. We have performed additional Shapiro-Wilk tests for normality on the normalized daytime/nighttime power change. Parametric and non-parametric tests were selected accordingly. This was added to the method section: “Shapiro’s normality test was performed on the normalized daytime/nighttime power change to determine the appropriate statistical tests, and the test statistics on change in power (difference between daytime and nighttime power) cannot conclude that the data is not normally distributed ($p > 0.05$), thus parametric tests were performed as described below.”

We conducted Shapiro-Wilk tests on each individual frequency band and concluded that we cannot reject the null hypothesis ($p > 0.05$). This indicates that non-parametric tests were not required, and we have updated the methods and results to use parametric student’s t-test in complementary to the parametric ANOVA. In parametric t-test, paired t-test (day vs night) is equivalent to 1-sample t-test against zero. We updated the manuscript to describe the test as: “A one-way analysis of variance (ANOVA) to compare across frequency bands revealed a significant effect of sensing frequency in GPi on daytime/nighttime power change ($F=18.22$, $p < 0.001$). Post-hoc analysis with Tukey’s test showed that the circadian rhythm in the alpha band showed more likely to increase at nighttime than low-beta ($q=-0.47$, $p=0.004$) and high-beta ($q=-1.09$, $p < 0.001$), and low-beta showed more increase at nighttime than high-beta ($q=0.63$, $p=0.003$) (**Figure 2C**). Comparison within each frequency band showed a significantly daytime/nighttime change in power was found in both low-beta (mean=0.38, $t=3.43$, $p_{\text{corrected}}=0.004$) and high-beta (mean=1.01, $t=9.08$, $p_{\text{corrected}} < 0.001$) with averaged positive values indicate that GPi power was most likely to decrease at night in both frequency band while alpha power was equally decreased or increased during nighttime, as indicated by a mean day/night change non-significantly different from 0 (mean=-0.04, $t=-0.36$, $p_{\text{corrected}}=1.0$)”.

10. Line 180: Provide the p-value for theta/alpha power.

We have updated the manuscript accordingly.

11. Line 195: The linear model is described as determining the effect of several variables on the “change in beta power”. More clarification is needed on what exactly the dependent variable is here – is it the magnitude of the difference in beta power between day and night? Or the direction of change (greater during the day or greater at night)? Or something else?

We have updated the Results and Methods section to clarify that the dependent variable is the normalized magnitude difference between day and night (continuous values) and not the direction (binary classification). “In particular, we investigated whether the change in beta power (dependent variable) between daytime and nighttime was significantly modulated by the following independent variables: the levodopa equivalent daily dose (LEDD)^{22,23}, the use of levodopa extended-release medication at nighttime, the use of non-dopaminergic medication at nighttime, the location of contact used for stimulation (ventral vs dorsal), the total electrical energy delivered (TEED), the Unified Parkinson's disease rating scale (UPDRS) total scores, and patient’s motor phenotype (tremor, intermediate, or akinetic subtype)”

Methods:

1. There is a discrepancy in the number of subjects reported between the Abstract and Introduction ($n = 119$) and the Methods ($n = 117$).

We have updated the manuscript accordingly.

2. Missing demographic data: fraction of unilateral vs bilateral, fraction of SenSight vs legacy lead, bipolar vs monopolar stimulation configurations.

We have added the following sentence in the results: “Subjects were implanted with the Medtronic Percept PC neurostimulator attached to unilateral (n = 38, 40.9%) or bilateral (n = 55, 59.1%) DBS leads in the GPi. Among them, 58 hemispheres (44.6%) were implanted with Medtronic 3387 quadripolar DBS electrodes, and 72 hemispheres (55.4%) were implanted with Medtronic SenSight segmented DBS electrodes.”

“All patients were programmed in a monopolar configuration and the two contacts adjacent to the contact(s) used for therapeutic stimulation were selected for bipolar chronic sensing.”

3. No clear description of which patient specific events were selected and how patients were advised to use those events

We have added the following sentence in the methods: “the Percept allows clinicians to set up event marking on the device, which allows subjects to mark specific events (e.g., ‘medication intake, ‘presence of dyskinesia’) that might be relevant for clinical care. Therefore, the time of each event marked during each recording was extracted and used as a surrogate for wakefulness. Although every subject was instructed to mark events whenever they occurred when out-of-clinic, the event markings were typically sparse and greatly varied across subjects.”

4. No description of how artifacts or missing data were handled in the Timeline data.

We have added the following sentence in the methods: “Data were pre-processed to handle potential outliers and missing data points. Outliers were defined as more than 5 standard deviations from the mean power recordings per subject. If outliers were detected, the values were removed and replaced by interpolating the previous and next data points. Recordings with missing data points were excluded. All spectral powers were z-score normalized using ± 12 hours of recordings to eliminate non-stationary slow-wave drifts and facilitate group analysis. All neural recordings were time-aligned on a 24-hour clock based on Eastern Daylight-Savings Time (EDT, GMT-04:00).”

5. Line 315: A definition (local field potential power) and appropriate units (μVp) should be provided for “brain recording”. This applies also to line 318, lines 341-348, and lines 356-357 (“neural signal”).

6. Line 317: The sensing-available configurations should be described, as not all readers may be familiar with this.

The Medtronic Percept device generates local field potential power without disclosing the actual algorithm, and the output are all proprietary values without units. We have updated the methods section to clarify this for readers not familiar with the recordings from embedded neurostimulators from Medtronic. We have added the following sentence in the methods: “This sensing-enable neurostimulator⁴¹ is capable of recording time-domain local field potentials (LFP) in-clinic (at 250Hz sampling rate) and neural power in preselected frequency band (bandwidth ± 2.5 Hz) every 10 minutes for up to 60 days. Patients include in this study had neuronal power recorded for at least 5 consecutive days. All of them were programmed in a monopolar stimulation (stimulation contact 1 or 2), allowing sensing from electrodes surrounding the stimulation contact to reduce stimulation artifacts⁴².”

7. Line 324: Either the method used to calculate the LEDD should be described, or a reference should be provided for the calculation.

The LEDD calculation is based on Tomlinson’s initial estimation in 2010 and further updated in 2023 by Jost et. al.

We have added the following sentence in the methods: “For each patient, the medications dose and schedule, were used to calculate the total LEDD based on conversion factor estimated by Tomlinson and colleagues^{24,25}”

Discussion:

1. Line 291: It is stated here that stimulation parameters (i.e., amplitude, pulse width, etc.) were not available in this dataset, but in the Methods it is reported that “therapeutic variables such as therapy frequency, therapy pulse width, and therapy amplitude were extracted” (line 366). These statements seem contradictory and should be reconciled.

We agree that our wording was confusing, and we have now modified it.

The stimulation parameters associated with each neural recordings are available and have been used to calculate the TEED for the generalized linear model.

Although our analysis shows no effect of stimulation on the LFP at a group level, change in stimulation parameters might alter LFP in individual patient, as it was shown in the original Supplementary Fig 1. However, in this study we have selected, for each patient, the first recording of 5 consecutive days with stable stimulation settings. Therefore, this dataset cannot be used to study the effect of varying stimulation parameters. Studies analyzing the effect of varying stimulation parameters within patients are needed. Although the impact of higher amplitude is interesting and require better understanding, it is not directly related to the results of this study. Thus, to improve clarity we have removed Supplementary Fig 1 from the paper.

Figures

Thank you so much for all the suggestions regarding the figures. We have updated the figures to include more details. Due to changes in the manuscript content by focusing on the GPI, we moved Figure 2 (STN) to supplementary figures.

1. For both Figures 1 and 2:

a. There’s no indication of what events were marked by the red-vertical lines – are they all the same, different event types

Events were created by clinicians for clinical purposes not related to the retrospective study and were thus different. Most commonly clinicians asked patient to indicate when taking medication, presence of tremor, dyskinesia, dystonia etc. In this study, these events were only used as surrogate markers of wakefulness in this study, thus we focused on the time of these events rather than the event itself. The list of events was added to the methods and the legend of the figure for clarity.

b. There is no scale marker on the polar plot radial axes

This has been modified accordingly.

c. It is not clear from the methods how their day / night algorithm would successfully differentiate between a ‘sleep period’ that in one patient LFP went up vs one that LFP went down when their algorithm appears to assume high activity periods are day/awake related (see major comment #3).

The algorithm described in the initial manuscript was used to divide the data into 2 categories, low and high power, irrespective of time of the recording and without any apriori selection. Times of event and time-of-day were then used to classify awake/asleep states.

However, we agree with reviewers’ s comments that this method is not optimal and adds complexity in data interpretation. Therefore, we have revised our method and updated it to a fixed time window at night (12am-5am) and during the day (3-8pm) (see our response to major comment #3 for full details).

d. There’s no description in the Results/Methods/Figure legend for how the plots in C were generated.

Does each row represent the average for all days recorded for one hemisphere? How did authors control differences in the number of days?

Each row in plot C represents the average power for all days recorded for one hemisphere and normalized with z-scores for across patients' comparison. Although the recordings were normalized, the number of days was not fully accounted for in the initial analysis. Therefore, we have modified the method and repeated the analysis selecting only the first 5 consecutive days of recording for each patient and the results remain the same with this consistent number of recorded days.

We have added a better description of these plots and modified the method, results, and figures to include the new analysis: “A population-level circadian heatmap was generated to visualize the normalized 24-hour power from all unique hemisphere recordings by sorting based on sensing frequency bands and the direction and magnitude of the effect of circadian rhythm to demonstrate the proportion of subjects with an increase of power in nighttime versus subjects with a decrease of power in nighttime.”

2. Figure 3 is labeled “Figure 2”. Additionally, would suggest using different symbols to denote different statistically significant findings and defining what each symbol means (eg, * is used twice in panel 3B, and ** is also used – what do these indicate?). Similarly, no explanation is given for the * symbol in panel 3D.

We have updated the figure and text accordingly: “(*) denotes $p < 0.05$, (**) denotes $p < 0.01$, and (***) denotes $p < 0.001$.”

3. Figure 4 is labeled as “Figure 3”, and is referred to in the text as “Figure 3C”.

We have updated the figure and text accordingly.

4. Table 2: Variables included in the linear model should be defined, particularly “channel” as this is not described in the text and it is unclear to what this is referring.

Channel refers to the contact pairs used for recording. We have updated the text and table to clarify this. In revised manuscript, we use the term “stimulation location” with dorsal and ventral for labeling. In the methods section, we also included that “The contact used for stimulation was defined as ventral 1-C+ vs dorsal 2-C+ (the location of the stimulating contact is provided in Supplementary figure 3).”

5. Supplementary Figure: The legend states that the circadian beta rhythm was strongly diminished by the increase in stimulation amplitude, but is it possible that this is just a floor effect? That is, the top panel clearly shows LFP power is overall reduced by the increase in stimulation amplitude, thus making the absolute difference between day and night smaller, but is the relative difference changed at all? Visually, it appears that there is still some rhythmicity to the LFP power.

We agree with the reviewer’s comment. There is still rhythmicity even with increased stimulation amplitude and the change might be due to floor effect. We have generated the circadian rhythm figure over the 2 time periods highlighted in Supplementary Figure 1 and displayed them in the figure below. The circadian rhythm is present as the reviewer pointed out, albeit significantly reduced. Although statistically we can conclude that a difference still exists, the effect shown in this case warrants further analysis for within-patient stimulation parameters’ effect on circadian rhythm in future perspective studies.

This figure was shown in the initial manuscript to make the reader aware of potential stim effects in individual patients. However, we have removed this figure from the manuscript and rephrased the paragraph on this topic (see minor concerns discussion #1 for more details).

Grammar/Syntax:

Line 111: Clarify “out stimulation settings through the recordings.”

Line 122: Change “reordered” to recorded.

Line 183: Change “Figure 3E” to Figure 3D.

Line 238: Change “(±2Hz)” to (±2.5Hz).

Line 323: Change “proxi” to proxy.

Table 1: Change “LEED” to LEDD.

The manuscript has been updated accordingly.

Reviewer #2 (Remarks to the Author):

In this study, Cagle et al reported a large retrospective cohort including 119 subjects with PD (165 hemispheres) with GPi and STN activity recorded chronically in the home environment. They find that both GPi and STN power are significantly modulated by circadian rhythm. Increased power was also found, especially in GPi (28%) at night, and both the frequency of the power band and the use of long-acting levodopa medications influenced the neurophysiological circadian rhythm. Overall, this is a well-powered study with the largest known subject numbers with chronic in-home recordings. However, some areas warrant constructive criticism and further consideration.

First, analyzing diurnal fluctuations associated basal ganglia oscillations has been conducted in previous papers^{1,2}. Sleep-related modulation in the pallidum has also been looked at in two recent studies^{3,4}. I take the major novelty of the work presented by the author as the large sample size included, and the finding that beta power during sleep could be higher than that during wakefulness (in 1/3 of the GPi recordings), an observation that has never been reported before. However, regarding this finding, a) the authors did not provide sufficient explanations for this phenomenon in the discussion, e.g., why beta power could be even higher during sleep than during wakefulness; what trait do these 1/3 patients have that differentiates them from the others? And b) the authors did not provide in-depth data supporting this finding and investigating what caused the beta enhancement, e.g., would it be in any case the beta elevation is caused by any type of artifact? Does beta burst have a longer duration during sleep for the beta-elevated group? During which sleep stage does the beta enhancement occur? Without more detailed information, it’s hard to judge the validity of the results and the implication of the observed beta enhancement at sleep.

I personally would suggest the authors compare the clinical and oscillatory features between the beta-increased and the beta-decreased groups to see what caused the difference. I would also suggest the authors record raw LFP data from at least several of these beta-increased patients to investigate how beta is changed during sleep-awake transitions. To the best of my knowledge, all published studies hold that beta power decreases from wakefulness to NREM sleep, and increases prior to awakening. If the presented results by the authors are valid, one possibility for the increase of beta during sleep could potentially be caused by the sleep spindle (11-15 Hz) and its higher-order harmonics^{5,6}. As it has been shown that sleep spindle exists in basal ganglia and could have an impact on the basal ganglia electrophysiological patterns. Another possibility could be that beta increased dramatically during REM sleep at night and in these beta-increase patients the proportion of REM sleep was exceptionally long. While in the daytime, beta was suppressed by medication and stimulation, resulting in the observation that beta increased at night. However, with only an average power every 10 min, it's hard to test these hypotheses.

Response:

We thank the reviewer for the comments and constructive feedback. As highlighted by the reviewers, one of the main novelties of this work is the increased beta power observed at night in some patients, particularly in GPi. However, it should also be pointed out that this is also the first report of GP activity recorded chronically in patient's home environment while on PD therapy (DBS and medications). Most studies have focused on STN except 2 recent publications studying GPi activity in the off-stimulation and off-medication state.

Although the reviewer's suggestions for further recordings are excellent, some, especially those requiring time domain LFP and sleep stage, we cannot be addressed with this retrospective dataset. The Percept only allows to record power of a selected frequency band that is averaged over every 10min. Patients were not using external sensors since they were under standard clinical care.

We agree that these results are surprising and have explored factors that might explain this phenomenon, including the effect of medications. In line with the reviewer's last point, our results studying the effect of medication suggest that beta power is more likely to increase at night in patients who do not take extended-release dopaminergic medication at night, which may be due to the medication wearing off and pathological beta reemerging at nighttime. To address the reviewer's comments, we have added a paragraph in the discussion speculating on the causes of increased beta power at night, including the potential contribution of sleep-spindle and/or an increased REM sleep stage time. This is also supported by the baseline spectral power comparison, showing that subjects who showed increased power at nighttime had significantly higher power in the awake off-medication off-stimulation state in in-clinic recordings.

Another important consideration is the potential contributions of sleep spindles, a typical electrophysiology feature (10-15Hz) that is present in electroencephalograms (EEG) during sleep, specifically the NREM phase^{30,31}. Sleep spindles are considered to be instrumental in memory consolidation and motor learning increases sleep spindle density³². In addition to the EEG studies, deep brain recordings also capture sleep spindles in both non-human primates³³ and humans³⁴. Although sleep spindles are primarily studied in the thalamus, given the thalamic-basal ganglia connection, it is possible that sleep spindles could affect chronic nighttime recordings in the GPi³⁵. Previous studies found that high frequency (≥ 130 Hz) DBS may have either an enhancing effect on sleep spindle density³⁶ or no significant effect on sleep spindle density between ON and OFF DBS states³⁷, but none reported a reduction of sleep spindles. In our study, chronic recordings in the low-beta band likely overlapped with the sleep spindle spectral ranges. Although, dopaminergic medication is known to reduce

beta power, it's effect on sleep spindles is not well understood. Therefore, it is possible that sleep spindles might contribute to the increase in power observed in some subjects. Future prospective studies are needed to fully address the presence of sleep spindles and their impact on chronic nighttime recordings.”

Although the current dataset does not allow to empirically fully explain this phenomenon, we believe that it is important to publish these new data collected chronically in patient's home environment and showing that GPi activity mainly decreases at night in the majority of and increase power in some patients. We are starting a prospective study using time domain data and polysomnography with the intention to answer questions about GPi activity (beta, beta bursts, etc.) related to sleep stages and to better understand this increased power. The results reported in the present study have helped to form the initial hypotheses for our prospective study and likely will inspire other ideas in the field.

The limitations of this retrospective study are now listed in the discussion:” It is important to highlight several limitations of our study. As a retrospective study, we did not have wearable sleep data or formal sleep diaries to confirm the sleep/wake states in all subjects and instead, we used a fixed time-of-day window for nighttime and daytime analysis and all findings were framed in the daytime/nighttime circadian rhythm, instead of sleep/awake cycle, similar to previous chronic STN circadian rhythm study¹⁴. Another limitation is the potential artifacts from recording in unrestricted naturalistic conditions. Cardiac artifact is commonly seen in Percept devices with legacy Medtronic electrodes³⁸, and we excluded sensing recordings from the delta/theta (7-8Hz) band to avoid contamination from electrocardiogram (ECG). Upper body motion can also lead to broadband motion artifacts³⁹ which might inflate the sensing power bands especially during the awake period. These artifacts may cause the recordings to display artificial power reduction at nighttime. Additionally, we did not have real-time measurements of the subjects' symptoms, disease states, or sleep stages at the time of recordings.”

We have analyzed in more details the potential effect of clinical features on power change at night and added the UPDRS scores and PD phenotype to the GLM analysis. We found that neither of these variables could explain the difference (increase vs decrease) in circadian rhythm. This analysis was added to the manuscript.

Second, the heat maps shown in Figure 1 and Figure 2 are somewhat misleading. If I understand correctly, the y-axis is the power of the recorded band but not necessarily the beta power, despite that the authors present examples of beta increase and decrease on the left panels. In fact, as the authors reported, in the GPi group, in 46.7% of the hemisphere the recorded frequency band is the theta/alpha band. It could be pointless to plot different frequency bands together and combinedly term them “recordings”, especially when the direction of modulation could oftentimes be opposite for different bands. I suggest plotting separate figures for each individual band.

Response:

As suggested, we have reorganized the circadian rhythm heatmap across patients according to their sensing frequency (divided into alpha, low beta, and high beta) and then sort by changes in spectral power.

Third, the automated circadian detection algorithm developed by the authors seems to be insecure to me. The authors used the time of ‘low’ and ‘high’ neuronal signal periods together with the event marked to determine day vs night. What if the patients had insomnia and could not get to sleep at night (strong beta power observed), but had too much daytime sleep (weak beta power observed) with the events representing only the time points that they were awakened to take medicine? Indeed, the beta power trace

shown in Figure 2a could probably be explained by this. In this scenario, the automated detection algorithm gives erroneous labels.

Response:

As pointed out by the first reviewer, we cannot exclude the possibility that some patients shown in Figure 2a might be awake at night and sleeping during the day, and the events might only be marked when patient is briefly awake during the daytime. However, we would then expect similar beta power when patient is awake, thus at the time of the event and at nighttime. In addition, we also examine the example patient's clinical visit time, and confirm that the recording shown as example starts during the clinical visit time (5pm EDT) and the patient would theoretically be awake during the visit and afterward when the power is low (awake, daytime).

However, we acknowledge that the method in the initial manuscript is not optimal thus based on reviewers' comments. We have updated our method to now use: 1) fixed time windows for nighttime (12am to 5am) and daytime (3pm-8pm), selected based on the assumption that patients will likely be asleep from 12am-5am and awake from 3pm-8pm; 2) using the raw signal (10-minute-averaged LFP power recorded on Percept PC neurostimulator) without any moving average low-pass filter. The results obtained with this new method are similar to those described in the original manuscript: the majority of GPi recordings showed a reduction of power at night while ~30% of the recordings showed an increased power at night. Similar observations were found with shorter windows from 12am-3am and 3pm-6pm, indicating that the results are robust to window size. Therefore, the manuscript has been modified to include this new method, and all figures have been updated. Additionally, the results emphasize differences in GP power at night vs day (instead of asleep vs awake).

Fourth, since the subthalamic and pallidal signals are influenced by the extended-release levodopa, the authors need to give more details about how they take this into consideration into their sleep/wake classification algorithm, e.g., showing the time of extended-release levodopa taken and how they influence the 24-hour spectrogram and the corresponding classification results.

Response:

Due to the concerns raised by multiple reviewers, after discussion among the authors in the manuscript, we have decided to revamp the analysis by not using the proposed automated time-frame selection algorithm, but instead using a fixed-time labeling approach for daytime and nighttime labels. All results were updated accordingly, and the method section was updated to reflect the new approach.

For the medication schedule, patients were either taking levodopa extended release (Levodopa ER) or levodopa immediate release (Levodopa) or both. For patients who are taking Levodopa ER only, they are taking the medication on a periodic schedule, with the last dosage right before sleep if more than 3 dosages were used per day. For patients who are taking both medications, they are taking the ER dosage right before sleep as instructed.

In result section, we included that "The use of nighttime medication is defined as specific pharmaceutical instruction of medication taken nightly or the last dose of periodic schedule that occur more than 3 times a day" We also updated the discussion to provide more details on our interpretation.

Other points:

1. Did the authors collect sleep related scales, e.g., PSQI, PDSS, RBDSQ, for the patients? Which could help explain the differential modulation of beta power between subjects

Response:

Unfortunately, sleep scales were not routine clinical data collection at our institute, so the scales suggested were not available in this retrospective dataset. We will include them in a prospective study.

2. The number of GPi implantations is much larger than the number of STN implantations. The authors should report how they choose between the two targets. Would this cause differences in clinical features between groups?

Response:

At the University of Florida, all patients underwent a 2-day screening process before an expert multidisciplinary committee decides whether they qualify as a candidate for DBS. At our center, patients are generally recommended GPi DBS when they report prominent levodopa-induced dyskinesia or when they exhibit concerning cognitive or mood difficulties. The STN target is usually recommended for patients with debilitating tremor, prominent akinesia, rigidity or those who are experiencing prominent dopaminergic medications adverse effects without significant cognitive impairment or dyskinesia, as described in Wong et al. 2020 (STN Versus GPi Deep Brain Stimulation for Action and Rest Tremor in Parkinson's Disease). It is therefore possible that differences across target might be explained by clinical features, which is why we do not compare STN vs GPi, and have moved STN results to the supplementary data.

However, we looked at the effect of symptoms on circadian rhythm in GPi. We retrieved the baseline UPDRS scores for all patients, computed their motor subtype (tremor-dominant, intermediate/akinetic, and postural instability and gait disability) based on multiple established criteria, and added it as a variable to the GLM models. However, the results indicated that PD motor subtype was not significantly associated with circadian rhythm.

3. The bipolar contact pair used for sensing was determined by the contact(s) used for continuous stimulation therapy. Could the author show the contact location of these patients, especially comparing the location between the beta increased and decrease patients.

Response:

We extracted the pre-operative T1 images and postoperative CT scans for all patients included in the study. We performed standard rigid co-registration between T1-MRI and CT images for lead localization. All T1 images were then normalized to MNI2009b standard brain template through 2-stage non-linear normalization (Affine + SyN through Advanced Normalization Tools ANTs). All sensing locations, as defined by the middle point between bipolar sensing contact pair, were colored by increased or decreased power at nighttime and shown below. No significant grouping or differentiation were observed.

4. In supplementary figure 1, I noticed that an increase in stimulation amplitude is also associated with a slight but consistent increase in the beta power during sleep. Given the y-axis is absolute power rather

than z score power, how do the authors explain this beta increase after turning up the stimulation amplitude?

Response:

We divided the data by stimulation amplitude and visualize the circadian rhythm for both data in the figure below. Nighttime power was not different between the two stimulation conditions (272.83 ± 16.34 vs. 258.40 ± 6.50 , $t = 0.68$, $p = 0.50$). However, the daytime power was significantly diminished. A larger reduction in Beta power was associated with more stimulation. Therefore, although still present, the circadian rhythm was not as consistent with 1.4mA compared to 0.5mA. In the revised manuscript, we have only included data with stable stimulation parameters. Therefore, this supplementary figure is not included in the revised manuscript.

5. In most sections, 119 subjects were mentioned, while in the Methods section it was 117 subjects. This discrepancy should be checked.

Response:

The manuscript has been updated accordingly.

6. There are two Figure 2 in the manuscript. The latter one should be Figure 3.

Response:

The manuscript has been updated accordingly.

References

1. Baumgartner, A., Hirt, L., Kern, D. & Thompson, J. Diurnal fluctuations of local field potentials follow sleep-wake behavior in Parkinson's disease. (2023). doi:10.21203/rs.3.rs-2468375/v1.
2. van Rheede, J. J. et al. Diurnal modulation of subthalamic beta oscillatory power in Parkinson's disease patients during deep brain stimulation. NPJ Parkinsons Dis 8, 88 (2022).
3. Yin, Z. et al. Pallidal activities during sleep and sleep decoding in dystonia, Huntington's, and Parkinson's disease. Neurobiology of Disease 182, 106143 (2023).
4. Yin, Z. et al. Pathological pallidal beta activity in Parkinson's disease is sustained during sleep and associated with sleep disturbance. Nat Commun 14, 5434 (2023).
5. Zhou, Y. et al. Methodological Considerations on the Use of Different Spectral Decomposition Algorithms to Study Hippocampal Rhythms. eNeuro 6, ENEURO.0142-19.2019 (2019).
6. Sushkova, O. S., Morozov, A. A. & Gabova, A. V. EEG Beta Wave Trains are not the Second Harmonic of Mu Wave Trains in Parkinson's Disease patients. in (2017).

Reviewer #3 (Remarks to the Author):

This study, by Cagle and Colleagues, comprises retrospective analysis of a very large data set recorded from people with Parkinson's disease that have been implanted with the Medtronic Percept device. The power of such recordings is that they allow spectral power at preset frequency bands to be collected at regular intervals for weeks and months, giving unique insight as to how these activities are modulated by circadian and diurnal rhythms. Studies using these data are still relatively rare and few have utilized the impressive numbers of patients reported here. In addition, the investigators report recordings from both STN and GPi. They also have self-reported measures of wakefulness, adding a measure of ground truth that has been absent from most other studies that have used these methods. The authors demonstrate that, as previously reported, in many patients “beta” power decreases during sleep. However, they show that this relationship is inverted with many patients and that the direction of the patients circadian modulation is dependent on the frequency of the beta power. This main observation is of potential importance to the development of closed-loop DBS approaches. However, there are currently several important caveats to the conclusions reached by the authors that considerably dilute its impact. Some of these issues are intrinsic to the methods used, for example that Percept data only allows one frequency band at a time to be analysed, while others could be addressed using further analysis.

Major Comments.

1. Are changes in diurnal beta profile driven by day time / waking differences or by night-time / sleep differences? The authors currently note two different diurnal profiles of beta, with their novel observation being that some show a relative increase during the night vs. the day and some show the opposite pattern. It is currently not addressed whether this is because of a day-time or night-time difference. A comparison of e.g. absolute power in comparable data sets (similar sensing frequencies) would be useful, to attempt to establish whether or not the novel finding represents increased beta during the night in some patients, or decreased beta during the day (or a mixture).

Response:

We thank the reviewer for the positive comments and the helpful suggestions. We have repeated the analysis with the raw spectral power instead of normalized to investigate. We extracted the nighttime power and daytime power from all patients with sensing frequency in beta band, and we performed a t-test between two groups of patients: those who had increased power at daytime and those who had increase power at nighttime. The results are also shown below. They indicate that daytime power was comparable between groups, but nighttime power was significantly different across groups.

This result is complementary to our extended-release dopaminergic medication analysis. Whether patients took extended-release dopaminergic medication or not, their daytime pathological beta would be suppressed due to frequent immediate-release dopaminergic medications and/or extended-released dopaminergic medications. However, nighttime pathological beta would only be controlled by extended-release dopaminergic medications, so patients who did not take extended-release dopaminergic medications may be more likely have significant increases in pathological beta at night.

In discussion, we have included that “In particular, a decrease in beta power at night was more likely observed in subjects treated with levodopa extended-release medications when compared to those who were not taking levodopa extended-release medications. The use of levodopa extended-release medications could suppress beta power at nighttime similar to the daytime²⁷⁻²⁹, thus masking the reemergence of pathological beta power at night as seen in subjects who were not taking extended-release medication.”

To determine whether the decrease versus increase power was due to a daytime or nighttime difference, we averaged the raw power over daytime (3-8pm) and nighttime (0-5am), and compared between patients with an increased power at nighttime versus those with a decreased power. We found a significant difference in power between daytime and nighttime within each subjects group (increased power, $W = 0$, $p_{\text{corrected}} < 0.001$ and decreased power $W = 0$, $p_{\text{corrected}} < 0.001$, paired Wilcoxon's sign-rank test). A significant difference between groups was also found but in the nighttime only (unpaired Mann-Whitney's U-test, $U = 809$, $p_{\text{corrected}} < 0.001$). Similar results were found for beta power band only. These results suggest that GPi power fluctuates with day/night cycles, and these fluctuations may be driven by a greater modulation of power at night than during the daytime.

2. How can the authors discriminate between beta oscillations and sleep spindles. In the data sets from GPi and STN, it is very rare (Figure 3A&C) to see a night-time increase unless the sensing frequency is below 18Hz. As the sensing frequency band is 2.5Hz on either side of the selected frequency, this is exactly where the sensed 'beta' frequency starts overlapping with the frequency of sleep spindles (9-16Hz). Sleep spindles can lead to large spectral peaks during sleep and do propagate through the basal ganglia (e.g. DOI: 10.1016/j.celrep.2022.111367) so it is highly likely that they are driving at least some of the activity observed in this frequency band. While the authors do not have polysomnographic data and this is a retrospective study, some work could be done to address this. One approach would be to revisit the analysis of extended release levodopa; sleep spindles would not be expected to be suppressed by L-DOPA to the same extent as Parkinsonian beta. Another would be to provide more granular analysis of the night-time profile of patients with higher night time beta vs lower night-time beta – In the example in figure 2, high beta in STN appears to have more of a 'peak' and less of a 'plateau'; is this representative of other STN or GPi data sets with high beta? This could be driven by sleep architecture (with spindles more common during some sleep stages, which are more common during certain parts of the night). Finally, addressing whether the changes in diurnal profile are driven by day time or night time 'beta' would also address this concern (similar absolute beta during the day but higher during the night would be expected if the effect is spindle-driven).

Response:

Sleep spindles are an important topic that we now discuss in the revised manuscript. Unfortunately, due to the lack of time-domain recordings, we cannot clearly identify the presence of sleep spindle in our

retrospective dataset. However, we would like to include our existing observations to facilitate the discussion and inform future prospective studies on out-of-clinic circadian rhythm.

Our existing analysis had identified that the use of nighttime extended-released levodopa contributes to some of the variability of the circadian rhythms. Using nighttime extended-released levodopa leads to more consistent decrease of power at nighttime when compared to daytime. Additional analysis shows that all but 2 patients taking levodopa ER at night had a decreased of power at nighttime, even within the range of sleep spindle. This observation suggests that the increased power at nighttime is possibly due to the reemergence of pathological beta activity, although the contribution of sleep spindles in the increase power cannot be ruled out.

This analysis was added to the manuscript. The lack of wearable or external sleep monitoring devices and possible role of sleep spindle have been added to the discussed.

3. The clinical phenotype of Parkinson's disease has a significant effect on the overall power of beta oscillations in the LFP. Are the patients with inverse circadian profiles more likely to be tremor-dominant, which is associated with less beta? This could suggest that the night-time beta is more likely to be sleep-spindle related and may help to address the questions in point 2.

Response:

We looked at the effect of symptoms on circadian rhythm in GPI. We retrieved the baseline UPDRS scores for all patients, computed their motor subtype (tremor-dominant, intermediate/akinetic, and postural instability and gait disability) based on multiple established criteria, and added it as a variable to the GLM models. However, the results indicated that PD motor subtype was not significantly associated with circadian rhythm.

4. A discussion of data quality / potential artifact sources is also warranted given known risk of

electrocardiographic and motion artifacts and the uncontrolled at-home nature of the recordings.

Response:

We have included discussion of potential motion artifacts (due to neck movement affecting the cables connecting DBS lead and implanted neurostimulator when awake, etc.) and how they may affect the signal collected via the chronic neural recording system. In discussion, we added that “Another limitation is the potential artifacts from recording in unrestricted naturalistic conditions. We excluded sensing recordings from the delta/theta (<8Hz) band to avoid any artifacts from electrocardiogram (ECG) commonly seen in Percept devices with legacy Medtronic electrodes³⁸. Upper body motion can also lead to broadband motion artifacts³⁹ which might inflate the sensing power bands, especially during the awake period. These artifacts may cause the recordings to display artificial power reduction at nighttime.”

Minor points.

1. Are profiles patient-specific? Where there is data from 2 hemispheres of a patient, do the hemispheres show a correlation in diurnal profile? Does this depend on the similarity/difference in sensing frequency between hemispheres? If a high-beta peak in a patient shows low beta during the night, can a low beta peak in that patient’s other hemisphere show an opposite circadian profile?

In our database, we found several patients with multiple frequencies tracked. Below are 2 examples of low beta power recorded in the same patient, same hemisphere, and showing opposite circadian rhythm. In this case, the 14.65Hz was increased at night while the 12.7Hz was reduced in nighttime. However, it should be noted that in this example, both the stimulation amplitude and medication were adjusted between these recording making it hard to draw any conclusion. Our data suggests that the direction of the change might not be patient specific but rather depends on the frequency sensed and the therapy (medication and/or stimulation). A study looking at the role of frequency, medication and stimulation on circadian rhythm within patient is needed to better understand this phenomena (not across patients like this study).

2. Introduction / discussion of canonical frequency bands. More effort should be made to discuss the meaning of activity in canonical frequency bands in the basal ganglia. The most obvious omission is the absence of a discussion of sleep spindles, which are an important potential confound. Other known changes in relevant basal ganglia activity bands could use some further discussion as well (e.g. differences between NREM/REM sleep stages, role during wakefulness, etc).

Response:

We have included more discussion on sleep stages: “Due to the lack of objective sleep stage measurements, we cannot provide a conclusion on the effect of circadian rhythms during specific stages of sleep in this retrospective study. However, our results generally match the observations from previous studies in the NREM stage. However, there was more variability within the alpha and low-beta bands. The increase in power at night, especially in low-beta power, could be explained by the large spectral window of the recording. Indeed, the neural power recorded chronically is set at a preselected frequency with a 5Hz window ($\pm 2.5\text{Hz}$). Therefore, it is possible that alpha power ($\sim 10\text{Hz}$) contaminated recordings at 13Hz.”

In addition, we also added sleep spindle discussion: “Another important consideration is the potential contributions of sleep spindles, a typical electrophysiology feature (10-15Hz) that is present in electroencephalograms (EEG) during sleep, specifically the NREM phase^{30,31}. Sleep spindles are considered to be instrumental in memory consolidation, and motor learning increases sleep spindle density³². In addition to the EEG studies, deep brain recordings have also captured sleep spindles in both non-human primates³³ and humans³⁴. Although sleep spindles have been primarily studied in the thalamus, given the thalamic-basal ganglia connection, it is possible that sleep spindles could affect chronic nighttime recordings in the GPi³⁵. Previous studies found that high frequency ($\geq 130\text{Hz}$) DBS may have either an enhancing effect on sleep spindle density³⁶ or no significant effect on sleep spindle density between ON and OFF DBS states³⁷, but none reported a reduction of sleep spindles. In our study, chronic recordings in the low-beta band likely overlapped with the sleep spindle spectral ranges. Although dopaminergic medication is known to reduce beta power, its effect on sleep spindles is not well understood. Therefore, it is possible that sleep spindles might contribute to the increase in power at night observed in some subjects. Future prospective studies are needed to fully address the presence of sleep spindles and their impact on chronic nighttime recordings.”

3. Figures 1C and 2C. Smoothing across patients (vertical axis) in figures 1C and 2C is not appropriate as it blends together independent patient data sets. Also, it would be much more informative to split these sorted profiles by canonical frequency band.

Response:

In the revised manuscript, we grouped the heatmap by canonical frequency bands. In addition, we removed the smoothing filter on the heatmap for more clear images for both GPi and STN.

4. Supplementary figure 1

There is a clear reduction in circadian profile when the stimulation is turned up. It is interesting to see that it is not only driven by daytime beta being reduced, but also by night-time beta being increased. Can the authors comment on potential explanations?

How much data is available on the effect of stimulation

Response:

As pointed by the reviewer, there is a clear reduction in circadian profile when the stimulation is turned up. For clarity, we have divided the data by stimulation amplitude and visualize the circadian rhythm for both amplitudes in the figure below. Statistically, nighttime power was similar in both condition at night (272.83 ± 16.34 vs. 258.40 ± 6.50 , $t = 0.68$, $p = 0.50$), however, the day power is significantly diminished with increased stimulation amplitude. This might be due to more therapeutic stimulation (1.5mA). However, in the revised manuscript, for each data (each hemisphere) stimulation parameters were thus the effect of stimulation within-patient cannot be studied. Therefore, this supplementary figure was not included in the revised manuscript. We do plan to study this effect in future study.

5. Methods - 'Day' vs 'night' period

Terminology is confusing; in figures 1 and 2 the authors call these periods 'sleep' and 'wake'. It is also not specified when the authors consider 'high neural signal' to mean day/wake or night/sleep – presumably this is dependent on the direction of the circadian profile? What is done for data sets where there were no significant differences between daytime and night-time? It should also be made a bit more explicit which analyses exactly these day/night or sleep/wake definitions are used for.

Response:

We understand the reviewer's concerns regarding our data-driven method which utilizes the differentiation of neural signals as the classifier and is a form of circular proof. We have updated our method to now use: 1) fixed time windows for nighttime (12am to 5am) and daytime (3pm-8pm), selected based on the assumption that patients will likely be asleep from 12am-5am and awake from 3pm-8pm; 2) using the raw signal (10-minute-averaged LFP power recorded on Percept PC neurostimulator) without any moving average low-pass filter. The results obtained with this new method are similar to those described in the original manuscript: the majority of GPi recordings showed a reduction of power at night while ~30% of the recordings showed an increased power at night. Similar observations were found with shorter windows from 12am-3am and 3pm-6pm, indicating that the results are robust to window size. Therefore, the manuscript has been modified to include this new method, and all figures have been updated. Additionally, the results emphasize differences in GP power at night vs day (instead of asleep vs awake).

6. The figure font sizes are often very small.

Response: We have revised the figures to increase the figure font sizes for easier reading.

7. This manuscript has many typos and needs a careful proofreading

Response: Thank you. We have carefully proofread and corrected any typos.

Reviewer #4 (Remarks to the Author):

I co-reviewed this manuscript with one of the reviewers who provided the listed reports as part of the Nature Communications initiative to facilitate training in peer review and appropriate recognition for co-reviewers.

REVIEWERS' COMMENTS

Reviewer #1 (Remarks to the Author):

Review of author responses to the comments I provided on the manuscript, titled “Chronic Intracranial recordings in the basal ganglia reveal patient specific circadian rhythms in Parkinson’s disease” as well as the revisions to the manuscript were thorough and thoughtful and substantively addressed all concerns raised.

In specific the revised conceptual framework – refining the focus to GPi recordings, critical evaluation and refinement of the day and night delineation, and enhancing the statistical analyses are commendable. It is evident that significant effort was put forth to not only address the points raised but also to enrich the overall quality and rigor of the work.

All major and minor concerns were more than adequately addressed. Figures are much improved and clear.

Reviewer #2 (Remarks to the Author):

I appreciate the significant revisions made by the authors, which have undoubtedly enhanced the quality of the manuscript.

However, as the authors were unable to meet my request for “recording raw LFP data from at least several patients with increased beta activity”, my primary concern, I cannot support the publication of this manuscript. I concur with the authors when they state, “The results reported in the present study have helped to form the initial hypotheses for our prospective study and will likely inspire other ideas in the field.” However, considering the broad readership of Nature Communications, a valid and thoroughly tested conclusion is a minimum requirement. I hope the authors understand that if these somewhat surprising results were obtained from rigorous prospective experimental designs, I would be more than happy to see it published in highly visible journals. Unfortunately, with the current methodologies, everything remains uncertain. To substantiate the conclusions, raw LFP data at the research level need to be recorded and analyzed.

Reviewer #3 (Remarks to the Author):

The authors have addressed my concerns.

Reviewer #4 (Remarks to the Author):

I co-reviewed this manuscript with one of the reviewers who provided the listed reports as part of the Nature Communications initiative to facilitate training in peer review and appropriate recognition for coreviewers.

REVIEWERS' COMMENTS

Reviewer #1 (Remarks to the Author):

Review of author responses to the comments I provided on the manuscript, titled “Chronic Intracranial recordings in the basal ganglia reveal patient specific circadian rhythms in Parkinson’s disease” as well as the revisions to the manuscript were thorough and thoughtful and substantively addressed all concerns raised.

In specific the revised conceptual framework – refining the focus to GPi recordings, critical evaluation and refinement of the day and night delineation, and enhancing the statistical analyses are commendable. It is evident that significant effort was put forth to not only address the points raised but also to enrich the overall quality and rigor of the work.

All major and minor concerns were more than adequately addressed. Figures are much improved and clear.

We thank the reviewer for the positive comment.

Reviewer #2 (Remarks to the Author):

I appreciate the significant revisions made by the authors, which have undoubtedly enhanced the quality of the manuscript.

However, as the authors were unable to meet my request for “recording raw LFP data from at least several patients with increased beta activity”, my primary concern, I cannot support the publication of this manuscript. I concur with the authors when they state, “The results reported in the present study have helped to form the initial hypotheses for our prospective study and will likely inspire other ideas in the field.” However, considering the broad readership of Nature Communications, a valid and thoroughly tested conclusion is a minimum requirement. I hope the authors understand that if these somewhat surprising results were obtained from rigorous prospective experimental designs, I would be more than happy to see it published in highly visible journals. Unfortunately, with the current methodologies, everything remains uncertain. To substantiate the conclusions, raw LFP data at the research level need to be recorded and analyzed.

We understand the reviewer’s concern. However, there is currently no FDA approved device that can record time domain data chronically at the subject’s home and we think that it is worth sharing the results of this manuscript with the scientific community.

However, this limitation is now acknowledged in the main manuscript.

The following paragraph was added to the discussion

“Finally, the temporal resolution of these recordings was limited to a preselected band power averaged over 10 minutes that prevented us from studying fast changes in GPi LFP signals that might have occurred at night and/or during the day.

Future clinical studies will need to better characterize the relationship between raw GPi LFP signals and awake/asleep states, sleep stages, PD symptoms, and related

parasomnias. These findings could be used to help guide the use of circadian rhythmicity in aDBS strategies.”

Reviewer #3 (Remarks to the Author):

The authors have addressed my concerns.
We thank the reviewer for the positive comment.

Reviewer #4 (Remarks to the Author):

I co-reviewed this manuscript with one of the reviewers who provided the listed reports as part of the Nature Communications initiative to facilitate training in peer review and appropriate recognition for coreviewers.
We thank the reviewer for the positive comment.